# From Information to Generative Exponent: Learning Rate Induces Phase Transitions in SGD

**Konstantinos Christopher Tsiolis**
University of Toronto & Vector Institute
kc.tsiolis@mail.utoronto.ca

**Alireza Mousavi-Hosseini**
University of Toronto & Vector Institute
mousavi@cs.toronto.edu

**Murat A. Erdogdu**
University of Toronto & Vector Institute
erdogdu@cs.toronto.edu

## Abstract

To understand feature learning dynamics in neural networks, recent theoretical works have focused on gradient-based learning of Gaussian single-index models, where the label is a nonlinear function of a latent one-dimensional projection of the input. While the sample complexity of online SGD is determined by the *information exponent* of the link function, recent works improved this by performing multiple gradient steps on the same sample with different learning rates — yielding a *non-correlational* update rule — and instead are limited by the (potentially much smaller) *generative exponent*. However, this picture is only valid when these learning rates are sufficiently large. In this paper, we characterize the relationship between learning rate(s) and sample complexity for a broad class of gradient-based algorithms that encapsulates both correlational and non-correlational updates. We demonstrate that, in certain cases, there is a phase transition from an "information exponent regime" with small learning rate to a "generative exponent regime" with large learning rate. Our framework covers prior analyses of one-pass SGD and SGD with batch reuse, while also introducing a new layer-wise training algorithm that leverages a two-timescales approach (via different learning rates for each layer) to go beyond correlational queries without reusing samples or modifying the loss from squared error. Our theoretical study demonstrates that the choice of learning rate is as important as the design of the algorithm in achieving statistical and computational efficiency.

## 1 Introduction

A key aspect of deep learning theory is to understand how neural networks can adapt to underlying data structure and achieve desirable statistical and computational complexity through their optimization dynamics. Towards this goal, several works have focused on learning target functions that depend on low-dimensional projections of data, such as single- and multi-index models. The complexity of learning these models depends on assumptions on the data distribution and on the optimization algorithm. For Gaussian data and online (also called one-pass) SGD on the squared loss, the number of training samples/iterations needed to learn a single-index model depends on a property of the target function known as the *information exponent* [BAGJ21]. Moreover, through computational lower bounds, the information exponent governs the complexity of any *Correlational Statistical Query* (CSQ) algorithm for learning single- and multi-index models [DNGL23, VE24]. Such an algorithm interacts with data only through queries of the form $yh(x)$ that lie within a fixed tolerance of their

expectation [Kea98, Rey20]. Though SGD can only heuristically be cast as CSQ, this formalism has served as a useful proxy to inform attempts to break this "curse of information exponent".

One such attempt is to consider variants of SGD that apply consecutive gradient updates — with distinct learning rates — on the same batch [DTA+24, LOSW24, ADK+24b]. This simulates more general non-correlational queries $h(x, y)$ and thus bears a connection to the broader class of *Statistical Query* (SQ) algorithms. Here, the complexity may be controlled by another property of the target function, namely the *generative exponent* [DPVLB24], which is at most as large as the information exponent, and can be significantly smaller. [JMS24] further demonstrated that batch reuse is not strictly necessary and other online algorithms can also break the curse of information exponent by instead choosing alternative loss functions, while leaving open the study of other components of the training algorithm.

A puzzling observation around reusing batches is that the sample complexity of full-batch gradient flow on the squared loss, through the best known upper bounds, still depends on the information exponent [BBSS22, MHWSE23]. This is in contrast with the intuition that reusing samples is sufficient for breaking out of CSQ, since full-batch gradient flow reuses the entire dataset at every "iteration". This observation suggests that the role of the learning rate, while ignored in the current literature, is also crucial in determining the query class and the resulting sample complexity of SGD.

## 1.1 Our Contributions

In this paper we aim to answer the following question:

*Can we characterize the regimes of complexity emerging from the choice of learning rate?*

We give a precise answer to the above question for a class of online iterative algorithms when learning single-index models. Specifically, we make the following contributions:

- In Section 3, we introduce our general framework and provide a careful learning-rate-dependent analysis of the sample complexity of learning single-index models, resulting in bounds that explicitly demonstrate phase transitions induced by the choice of learning rate hyperparameters.
- We show that our framework is expressive enough to capture both vanilla online SGD (Section 4.1) and algorithms with non-correlational update rules such as SGD with batch reuse (Section 4.2). For the latter, our analysis interpolates between the complexity $n = \Theta(d^{(p_* - 1) \vee 1})$ and the online SGD complexity $n = \Theta(d^{(p-1) \vee 1})$ as the learning rates decrease, where $p$ and $p_*$ are the information and generative exponents respectively, $n$ is the number of training samples, and $d$ is in the input dimension. Specifically, we prove phase transitions in the complexity as a function of the learning rate for the first update on each batch.
- In Section 4.3, we show that even when considering squared loss, batch reuse is not the only approach that goes beyond CSQ limitations. In particular, we introduce a new layer-wise training algorithm that uses a different scaling of learning rate for the first and second layers of the network, thus using a two-timescales dynamics. We demonstrate that the performance of this algorithm also depends critically on the learning rate of the second layer. When the latter is sufficiently large, the algorithm can recover the target with almost linear sample complexity when the square of the link function has information exponent 1 or 2. Additionally, this analysis can be extended to a sparsely-connected network with $D$ layers, with the same conclusion holding (under further assumptions) when the $D$th power of the link function has information exponent 1 or 2.

The rest of the paper is organized as follows. We provide background on Gaussian single-index models in Section 2. In Section 3, we introduce a generic framework to study online gradient-based algorithms and provide our main result. We instantiate this framework for SGD with batch-reuse and the layer-wise two-timescales algorithm in Section 4. We sketch the proof of our main result in Section 5, and we conclude in Section 6.

**Notation.** For $k \in \mathbb{N}$, we use $[k]$ to denote the set $\{1, \ldots, k\}$. All asymptotic notation is with respect to the input dimension $d$. We use $\tilde{O}(\cdot)$ and $\tilde{\Theta}(\cdot)$ to denote $O(\cdot)$ and $\Theta(\cdot)$ up to polylogarithmic

factors, respectively. Similarly, the relations $\lesssim$ and $\gtrsim$ denote bounds up to polylogarithmic factors. We write $a \asymp b$ when $a \lesssim b$ and $a \gtrsim b$. An event is said to occur *with high probability* if its probability is at least $1 - o_d(1)$. The notations $\langle \cdot, \cdot \rangle$ and $|| \cdot ||$ refer respectively to the Euclidean inner product and norm for vectors in $\mathbb{R}^d$ in the absence of a subscript, while $||\boldsymbol{v}||_{\boldsymbol{A}} = \boldsymbol{v}^\top \boldsymbol{A} \boldsymbol{v}$ for $\boldsymbol{v} \in \mathbb{R}^d$ and $\boldsymbol{A} \in \mathbb{R}^{d \times d}$. For $\boldsymbol{w} \in \mathbb{S}^{d-1}$, $\boldsymbol{P}_{\boldsymbol{w}}^\perp$ denotes the projection onto the tangent space of $\mathbb{S}^{d-1}$ at $\boldsymbol{w}$, i.e., $\boldsymbol{P}_{\boldsymbol{w}}^\perp = \boldsymbol{I}_d - \boldsymbol{w}\boldsymbol{w}^\top$. For any $g \in L^2(\mathcal{N}(0,1))$, we write its Hermite expansion as $g(z) = \sum_{k=0}^\infty u_k(g)\mathsf{He}_k(z)$, where $\mathsf{He}_k$ denotes the $k$-th probabilist's Hermite polynomial [O'D21] and $u_k(g) = \mathbb{E}_{z \sim \mathcal{N}(0,1)}[g(z)\mathsf{He}_k(z)]$ is the $k$-th Hermite coefficient of $g$.

## 1.2 Related Work

**Feature Learning and Single-Index Models.** There is a vast body of literature on algorithms for learning Gaussian single-index models, see e.g. [DH18, CM20]. Here, we focus on more recent works that use gradient-based training. [BAGJ21] studied online SGD for learning high-dimensional single-index models with known non-linearity, where they introduced the information exponent as the quantity controlling the number of samples needed to learn the model. The representation learned by a network on single-index models with information exponent 1 was studied in [BES+22, MHWSE23], while [BBSS22] considered gradient flow for learning functions with higher information exponent. On multi-index models, [DLS22] considered one gradient step for learning polynomials, and [AAM23] studied learning general multi-index models where a saddle-to-saddle dynamics can

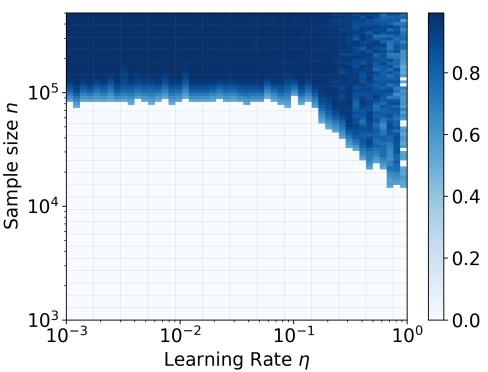

Figure 1: Combinations of learning rate $\eta$ and sample size $n$ which achieve alignment $\langle \boldsymbol{w}, \boldsymbol{\theta}_* \rangle \geq 0.5$ for a network (2.2) with $N = 1$ trained by alternating SGD (Algorithm 2) in the setting $\sigma_* = \sigma = \mathsf{He}_3$ and $d = 50$.

emerge. General multi-index models remain difficult to analyze [DKL+23, BBPV23, MHWE25]. However, several works have studied the simpler case of additive models [OSSW24, RL24, ŞBH24, RNWL25, BAEVW25].

CSQ and SQ lower bounds for learning single-index models where developed in [DLS22] and [DPVLB24] respectively, where the former depends on the information and the latter depends on the generative exponent. Similar lower bounds were derived in [AAM23] for multi-index models, where the "leap exponent" controls the complexity, and [TDD+24] studied approximate message passing as a proxy for computational lower bounds.

Going beyond unstructured isotropic Gaussian data, [JKMS25] recently studied single-index models with general spherically symmetric input distributions, positing an SQ lower bound for running time (which is attained by an SGD variant) and a low-degree polynomial (LDP) sample complexity lower bound. Many works considered the existence of additional input structure or modifications of the single-index model, such as a spiked covariance [MHWSE23, BES+23, BG24, BQI25, JMJS25, ZMN+25], sparsity in the input [VE24], or a perturbation of the target [CMM25]. The recovery of the low-dimensional multi-index subspace has been used to go beyond standard learning frameworks, e.g., to obtain better theoretical guarantees for adversarial robustness [MHJE25].

**Learning Rate and Generalization.** Numerous works have studied the effect of learning rate on optimization and generalization in deep learning. Notably, deep networks with large learning rate can operate near the "edge of stability" [CKL+21], where it has been empirically observed that such large learning rates improve generalization by preferring flat minima [LWM19, LBD+20, JAA+21, BD21, and references therein], learning sparse features [AVPVF23], or obtaining larger margin [CWM+24]. Closer to our setting, [ADK+24a] study the optimal choice of learning rate for online SGD. However, while their algorithm always remains in a correlational regime, we consider a wide range of learning rates to understand the effect of non-optimal choices in practice, and demonstrate phase transitions in the behavior of the SGD depending on stepsize, going from correlational regimes dominated by information exponent to full statistical query regimes dominated by generative exponent.

## 2 Problem Setup

We consider a supervised regression setting where the inputs are drawn from the standard Gaussian distribution and the labels are generated according to the *single-index model*, i.e.

$$y_i = \sigma_*(\langle \boldsymbol{x}_i, \boldsymbol{\theta}_* \rangle) + \zeta_i, \quad \boldsymbol{x}_i \overset{i.i.d.}{\sim} \mathcal{N}(\boldsymbol{0}, \boldsymbol{I}_d), \tag{2.1}$$

where $\boldsymbol{\theta}_* \in \mathbb{S}^{d-1}$ is the ground truth direction, $\sigma_* : \mathbb{R} \to \mathbb{R}$ is a (nonlinear) link function, and $\zeta_i$ is i.i.d. symmetric sub-Weibull[1] [VGNA20] label noise with $O(1)$ tail parameter.

We learn the above model with a two-layer neural network[2] $f$ with $N$ hidden neurons, first-layer weights $\boldsymbol{w}_j \in \mathbb{S}^{d-1}$, biases $b_j \in \mathbb{R}$, second layer weights $a_j \in \mathbb{R}$, and polynomial activations $\sigma_j : \mathbb{R} \to \mathbb{R}$ as in [LOSW24]. When convenient, we use the shorthand of encoding the first layer weights as rows in the matrix $\boldsymbol{W} \in \mathbb{R}^{N \times d}$, the second layer weights by the vector $\boldsymbol{a} \in \mathbb{R}^N$, and the biases by the vector $\boldsymbol{b} \in \mathbb{R}^N$. The network outputs a weighted average of the hidden layer activations:

$$f(\boldsymbol{x}; \boldsymbol{W}, \boldsymbol{a}, \boldsymbol{b}) = \frac{1}{N} \sum_{j=1}^{N} a_j \sigma_j(\langle \boldsymbol{x}, \boldsymbol{w}_j \rangle + b_j). \tag{2.2}$$

Our objective is to characterize the number of iterations (and thus, the number of samples) required for *weak recovery* of $\boldsymbol{\theta}_*$ as a function of the learning rate for online iterative algorithms. That is, starting from a uniform initialization on the sphere $\mathbb{S}^{d-1}$ where $\langle \boldsymbol{\theta}, \boldsymbol{w}_j^{(0)} \rangle \asymp d^{-1/2}$ with high probability, we seek $T$ such that $\langle \boldsymbol{\theta}_*, \boldsymbol{w}_j^{(T)} \rangle \gtrsim 1/\operatorname{polylog} d$. Studies of online SGD and variants [BAGJ21, DTA+24, LOSW24] argue that achieving weak recovery is the computational bottleneck in fitting a single-index target. Once this is achieved, *strong recovery* (i.e., $\langle \boldsymbol{\theta}_*, \boldsymbol{w} \rangle \geq 1 - \varepsilon$ for some $\varepsilon > 0$) and approximation of the target via ridge regression on $\boldsymbol{a}$ proceed with $\tilde{\Theta}(d)$ sample complexity. We extend these findings to our general class of gradient-based algorithms in Appendices B.6 and B.7.

We introduce two properties of $\sigma_*$ that are known to control the complexity of gradient-based learning and the complexity of learning with statistical queries.

**Definition 2.1** (Information Exponent, [BAGJ21])**.** *For any $g \in L^2(\mathcal{N}(0,1))$, let $u_k(g)$ denote the $k$th coefficient in its Hermite expansion. The information exponent of $g$ is defined as*

$$\mathrm{IE}(g) := \min\{k > 0 : u_k(g) \neq 0\}. \tag{2.3}$$

Throughout this paper, we denote the information exponent of the link function $\sigma_*$ in (2.1) by $p$, and we use the notation $p_i := \mathrm{IE}(\sigma_*^i)$ for $i \geq 2$ to denote the information exponents of powers of $\sigma_*$. [BAGJ21] show that online SGD with the square loss has sample complexity $n = \tilde{\Theta}(d^{(p-1)\vee 1})$, while [DNGL23] introduces smoothed online SGD, which achieves the optimal $n \gtrsim d^{(p/2)\vee 1}$ sample complexity for the class of CSQ learners. Beyond correlational queries, the generative exponent controls the complexity of any statistical query learner.

**Definition 2.2** (Generative Exponent, [DPVLB24])**.** *For any $g \in L^2(\mathcal{N}(0,1))$, the generative exponent is defined as the smallest information exponent over all $L^2$ transformations of $g$, i.e.,*

$$\mathrm{GE}(g) = \inf_{\mathcal{T} \in L^2(g_\# \mathcal{N}(0,1))} \mathrm{IE}(\mathcal{T}g). \tag{2.4}$$

Note that $\mathrm{GE}(g) \leq \mathrm{IE}(g)$ for all $g$. Throughout this paper, we denote the generative exponent of $\sigma_*$ in (2.1) by $p_*$. While [DPVLB24] developed an optimal algorithm with sample complexity $n \gtrsim d^{(p_*/2)\vee 1}$, [LOSW24, ADK+24b] showed that SGD has sample complexity $n \gtrsim d^{(p_*-1)\vee 1}$ when going over each sample twice. Both leverage the following crucial property.

**Lemma 2.3** ([LOSW24] Proposition 6, Lemma 8)**.** *Suppose there exists an orthonormal polynomial basis of the space $L^2((\sigma_*)_\# \mathcal{N}(0,1))$. Then there exists $I \in \mathbb{N}$ such that $\mathrm{IE}(\sigma_*^I) = p_*$. Moreover, if $\sigma_*$ is polynomial of degree at most $q$, then $I \leq C_q$ for some constant $C_q$ depending only on $q$.*

---

[1]Note that the class of sub-Weibull random variables includes sub-Gaussian and sub-exponential random variables and is closed under transformations with at most polynomial growth (up to changing the tail parameter).

[2]In Appendix C.4, we propose an algorithm for a network with $D > 2$ layers and prove sample complexity improvements over the batch reuse SGD of [ADK+24b, LOSW24] under assumptions on $\sigma_*$ and the $\sigma_j$.

This result can be used to develop optimal algorithms [CWL+25], and has two major implications. First, it immediately implies that batch reuse SGD — *with the correct choice of hyperparameters* — has linear sample complexity (up to log factors) for any polynomial target. Second, and importantly for our work, it suggests that if the update for $\boldsymbol{w}_j$ contains the monomial transformation $y^I = (\sigma_*(\langle \boldsymbol{x}, \boldsymbol{\theta}_* \rangle))^I$, the sample complexity can be reduced to depend on $p_*$ instead of $p$. In the subsequent sections, we emphasize that this will only occur if the scaling of the $y^I$ term (as determined by the learning rates) is sufficiently large. Otherwise, we fall back into the information exponent regime.

## 3   Sample Complexity of a Generic Online Algorithm

To generalize several notions of gradient-based learning of single-index models, we consider updates to a first-layer weight[3] $\boldsymbol{w}$ of the form

$$\boldsymbol{w}^{(t+1)} \leftarrow \boldsymbol{w}^{(t)} + \gamma \psi_\eta(y^{(t)}, \langle \boldsymbol{x}^{(t)}, \boldsymbol{w}^{(t)} \rangle) \boldsymbol{P}^\perp_{\boldsymbol{w}^{(t)}} \boldsymbol{x}^{(t)}, \quad \boldsymbol{w}^{(t+1)} \leftarrow \frac{\boldsymbol{w}^{(t+1)}}{\|\boldsymbol{w}^{(t+1)}\|}, \qquad (3.1)$$

where $(\boldsymbol{x}^{(t)}, y^{(t)})$ is an i.i.d. draw from the target single-index model (2.1), $\gamma > 0$, $\eta \geq 0$, $\boldsymbol{P}^\perp_{\boldsymbol{w}^{(t)}} = \boldsymbol{I}_d - \boldsymbol{w}\boldsymbol{w}^\top$ (i.e., the projection onto the tangent space of the unit sphere at $\boldsymbol{w}$), and $\psi_\eta$ is an update function based on a "general gradient oracle". This formulation is similar to that of [CWL+25], who also use generalized gradients, but additionally incorporate weight perturbation and averaging to achieve optimal rates. We view latter modification as complementary to our work, but we note that it can push the algorithms we study in Section 4 towards SQ-optimality *when $\gamma$ and $\eta$ are both chosen as large as possible*. The use of a spherical dynamics is common in studies of gradient-based learning of single-index models and is motivated by the fact that the optimization is done over the unit sphere $\mathbb{S}^{d-1}$ [BAGJ21, DNGL23, LOSW24, ADK+24b].

Note that due to the rotational symmetry of the Gaussian single-index model, the standard squared loss $\ell(y, y') = (y - y')^2$ and the *correlation loss* $\ell(y, y') = 1 - yy'$ induce identical gradients in the population limit, and are known to produce similar dynamics under small initialization of the second layer (see e.g., [AAM23, LOSW24]). Therefore, in this paper we examine choices of the oracle $\psi_\eta$ based on the correlation loss.

In the examples we consider, $\gamma$ and $\eta$ are learning rates that arise due to multiple gradient updates within the same iteration. Each plays a distinct role, with $\gamma$ serving as a "global" learning rate, and $\eta$ controlling the scale of any non-correlational terms in the oracle $\psi$. Both of these learning rates directly influence sample complexity, but *only $\eta$* will induce the aforementioned phase transition of interest. Moreover, the largest possible value of $\gamma$ is constrained by the value of $\eta$ (otherwise the algorithm can diverge).

We view such algorithms as inherently online, where the first step performs some transformation of the labels (modulated by $\eta$), and the second step uses the update of (3.1). This view can be extended to any constant number of steps on the same batch of samples. For such algorithms, the dependence of $\psi_\eta$ on $\eta$ becomes nonlinear, and the key quantities elucidating the effect of this hyperparameter on the sample complexity are the Hermite coefficients[4]

$$\mu_i(\eta) := \mathbb{E}_{(a,b) \sim \mathcal{N}(\boldsymbol{0}, \boldsymbol{I}_2)}\big[\psi_\eta\big(\sigma_*(a), b\big) \mathsf{He}_i(a) \mathsf{He}_{i-1}(b)\big], \quad i \in [r]. \qquad (3.2)$$

We highlight three examples in the next section:

1. **Online SGD**: $\psi_\eta(y, z) = y\sigma'(z)$ and $\mu_i = iu_i(\sigma_*)u_i(\sigma)$. There is no dependence on $\eta$ as each iteration involves a single gradient step with learning rate $\gamma$. See Section 4.1 for details.

2. **Batch reuse SGD**: $\psi_\eta(y, z) = y\sigma'(z) + \sum_{k=2}^{\deg(\sigma)} \frac{(\eta d)^{k-1}(\sigma^{(k)}(z))(\sigma'(z))^{k-1}}{(k-1)!} y^k$ and $\mu_i(\eta) \asymp \sum_{k=1}^r \frac{(\eta d)^{k-1}}{(k-1)!} u_{i-1}(\sigma^{(k)}(\sigma')^{k-1}) u_i(\sigma_*^k)$. The algorithm first takes a gradient step with learning rate $\eta$, followed by another gradient step on the same batch with learning rate $\gamma$. This algorithm was previously studied in [ADK+24b, LOSW24]. See Section 4.2 for details.

---

[3]In what follows, we drop the subscript $j$ for convenience.

[4]The formulation below is valid in the noiseless case. We handle sub-Weibull label noise in Appendix B.

3. **Alternating SGD**: $\psi_\eta(y, z) = y\sigma'(z) + \eta y^2 \sigma(z)\sigma'(z)$ and $\mu_i(\eta) = iu_i(\sigma_*)u_i(\sigma) + \eta u_{i-1}(\sigma\sigma')u_i(\sigma_*^2)$. The algorithm first takes a gradient step on the second layer with learning rate $\eta$, followed by a gradient step on the first layer with learning rate $\gamma$. This is our novel variant that we detail in Section 4.3.

We make the following assumptions on the target link function $\sigma_*$ and the student update $\psi_\eta$. The first ensures that all noise terms are sub-Weibull, allowing us to make concentration arguments. Prior works make a similar assumption (c.f., [ADK$^+$24a, Assumption 1], [LOSW24, Assumption 2]).

**Assumption 3.1.** *The link function $\sigma_*$ has at most polynomial growth, i.e., there exist constants $K_1, K_2 > 0$ such that $|\sigma_*(z)| \leq K_1(1+|z|)^{K_2}$ for all $z \in \mathbb{R}$. The update function $\psi_\eta$ is a polynomial of degree at most $r = \Theta(1)$ in each of its arguments and with $O(1)$ coefficients.*

The second assumption provides some degree of alignment between $\sigma$ and $\sigma_*$, without which the model misspecification is so severe that weak recovery may not be achieved (c.f., [BAGJ21, Remark 2.3], [LOSW24, Assumptions 2 and 3], [ADK$^+$24b, Assumption 4], [CWL$^+$25, Assumption 4.1(b)]). We explore how it manifests for different examples in Section 4.

**Assumption 3.2.** *For any $i^* \in \underset{\substack{1 \leq i \leq r \\ \mu_i \neq 0}}{\operatorname{argmin}} |\mu_i(\eta)|^{-1}(d^{\frac{i-2}{2} \vee 0})$, we have $\mu_{i^*}(\eta) > 0$.*

Below, we state our main result for a generic gradient-based algorithm.

**Theorem 3.3.** *Suppose Assumptions 3.1 and 3.2 hold. Let $\boldsymbol{w}^{(0)} \in \mathbb{S}^{d-1}$ such that $\langle \boldsymbol{\theta}_*, \boldsymbol{w}^{(0)} \rangle \asymp d^{-1/2}$. Then, there exists $C \gtrsim 1/\operatorname{polylog} d$ such that for any $\delta \in (0,1)$, if $\gamma \leq C\delta \max_{1 \leq i \leq r} \mu_i d^{-(\frac{i}{2} \vee 1)}$, then*

$$T(\eta) = \min_{\substack{1 \leq i \leq r \\ \mu_i > 0}} \tilde{\Theta}\big(\gamma^{-1}(\mu_i(\eta))^{-1} d^{\frac{i-2}{2} \vee 0}\big) \tag{3.3}$$

*iterations of (3.1) are necessary and sufficient to achieve $\langle \boldsymbol{\theta}_*, \boldsymbol{w} \rangle \gtrsim 1/\operatorname{polylog} d$ with probability at least $1 - \delta$.*

The nonsmooth $\min$ operation in (3.3) implies that the sample complexity $T(\eta)$ can exhibit nonsmooth phase transitions when the index yielding the smallest term changes. We can identify these phase transitions by determining for which $\eta$ we have $\mu_i^{-1}(\eta)d^{\frac{i-2}{2}} = \mu_j^{-1}(\eta)d^{\frac{j-2}{2}}$ for given indices $i \neq j$. This is the phenomenon we will concretely illustrate with batch reuse SGD and alternating SGD in Section 4, where the coefficients $\mu_i$ depend on the information and generative exponents of $\sigma_*$ and are non-decreasing in $\eta$.

**Remark 3.4** (On the Optimal Choices of $\gamma$ and $\eta$). *The optimal choice of $\gamma$ depends on $\eta$ through the coefficients $\mu_i(\eta)$. It is immediate from the statement of Theorem 3.3 that the best choice is $\gamma \asymp \max_{1 \leq i \leq r} \mu_i(\eta)d^{-(\frac{i}{2} \vee 1)}$. Then, the sample complexity reads*

$$T(\eta) = \min_{\substack{1 \leq i \leq r \\ \mu_i > 0}} \tilde{\Theta}\big((\mu_i(\eta))^{-2} d^{(i-1)\vee 1}\big). \tag{3.4}$$

*When the $\mu_i$ are all non-decreasing in $\eta$, it is clear that the best sample complexity is achieved by taking $\eta$ as large as possible subject to the constraint imposed by Assumption 3.1.*

## 4 Examples

We illustrate the applicability of Theorem 3.3 with three example algorithms: online SGD, batch reuse SGD, and a layer-wise algorithm we term *alternating SGD*. The latter two algorithms contain non-correlational terms that, when $\eta$ is chosen sufficiently large, lead to strictly better sample complexity than vanilla online SGD. We outline the computation of $\mu_i$ in each of these cases to explicitly capture this dependence on $\eta$. We assume that the largest possible $\gamma$ is chosen for each algorithm (recall from Theorem 3.3 that $\gamma \lesssim \max_{1 \leq i \leq r} \mu_i(\eta)d^{-\frac{i}{2} \vee 1}$).

## 4.1 Online SGD

We study vanilla (spherical) online SGD on the correlation loss $\ell(y, y') = 1 - yy'$ as a baseline for our framework. When $a_j = 1$, it gives the one-step update

$$\boldsymbol{w}_j^{(t+1)} \leftarrow \boldsymbol{w}_j^{(t)} + \gamma y^{(t)} \sigma'(\langle \boldsymbol{x}^{(t)}, \boldsymbol{w}_j^{(t)} \rangle) \boldsymbol{P}_{\boldsymbol{w}_j^{(t)}}^\perp \boldsymbol{x}^{(t)}, \quad \boldsymbol{w}_j^{(t+1)} \leftarrow \frac{\boldsymbol{w}_j^{(t+1)}}{||\boldsymbol{w}_j^{(t+1)}||} \tag{4.1}$$

for $j \in [N]$. The update oracle $\psi_\eta(y, z) = y\sigma'(z)$ is a single correlational term that does not depend on $\eta$. Consequently, there is only one sample complexity regime and no phase transition. A short calculation in Appendix C.1 shows that $\mu_i = i u_i(\sigma_*) u_i(\sigma)$. With this in hand, Theorem 3.3 gives the following result.

**Corollary 4.1.** *Assume $u_p(\sigma_*)u_p(\sigma) > 0$, $\gamma \asymp d^{-(\frac{p}{2} \vee 1)}$, and $\boldsymbol{w}^{(0)} \in \mathbb{S}^{d-1}$ such that $\langle \boldsymbol{\theta}_*, \boldsymbol{w}^{(0)} \rangle \asymp d^{-1/2}$. Then, with high probability, $\tilde{\Theta}(d^{(p-1)\vee 1})$ iterations of the update (4.1) are necessary and sufficient to achieve weak recovery.*

This matches both the sample complexity bound and the constraint on $\gamma$ in [BAGJ21, Theorem 1.3].

## 4.2 Batch Reuse SGD

Next, we consider the modification to online SGD where two gradient steps are taken on the same data. [DTA+24] employ dynamical mean field theory (DMFT) to argue this enlarges the class of targets learnable with linear sample complexity. This is because the two-step update implicitly introduces a nonlinear label transformation (and thus, non-correlational terms) into the update that can speed up learning. This approach — which we detail in Algorithm 1 and call *batch reuse SGD* — was further studied in [LOSW24] and [ADK+24b], where its sample complexity was characterized for any polynomial target, regardless of whether it can be learned in linear time. Both employ two distinct learning rates, which is necessary to simultaneously control the normalization error and ensure that the non-correlational term is sufficiently large (see in particular [LOSW24, Section 4.2]).

---

**Algorithm 1:** Batch Reuse SGD

**Input:** Learning rates $\eta, \gamma > 0$, sample size $T$
**Initialize** $\boldsymbol{w}^{(0)} \sim \text{Unif}(\mathbb{S}^{d-1})$
**for** $t = 0$ *to* $T - 1$ **do**
    Draw i.i.d. sample $(\boldsymbol{x}, \boldsymbol{y})$
    $\tilde{\boldsymbol{w}}^{(t)} \leftarrow \boldsymbol{w}^{(t)} + \eta y \sigma'(\langle \boldsymbol{x}, \boldsymbol{w}^{(t)} \rangle) \boldsymbol{P}_{\boldsymbol{w}^{(t)}}^\perp \boldsymbol{x}$
    $\boldsymbol{w}^{(t+1)} \leftarrow \boldsymbol{w}^{(t)} + \gamma y \sigma'(\langle \boldsymbol{x}, \tilde{\boldsymbol{w}}^{(t)} \rangle) \boldsymbol{P}_{\boldsymbol{w}^{(t)}}^\perp \boldsymbol{x}$
    Normalize $\boldsymbol{w}^{(t+1)} \leftarrow \boldsymbol{w}^{(t+1)} / ||\boldsymbol{w}^{(t+1)}||$
**end**
**Output** $\boldsymbol{w}^{(T)}$

---

Combining the two update steps for $\boldsymbol{w}$ in Algorithm 1 gives (before normalization)

$$\boldsymbol{w}^{(t+1)} = \boldsymbol{w}^{(t)} + \gamma y \sigma'\big(\langle \boldsymbol{x}, \boldsymbol{w}^{(t)} \rangle + \eta ||\boldsymbol{x}||_{\boldsymbol{P}_{\boldsymbol{w}^{(t)}}^\perp}^2 y \sigma'(\langle \boldsymbol{x}, \boldsymbol{w}^{(t)} \rangle)\big) \boldsymbol{P}_{\boldsymbol{w}^{(t)}}^\perp \boldsymbol{x}. \tag{4.2}$$

The norm $||\boldsymbol{x}||_{\boldsymbol{P}_{\boldsymbol{w}^{(t)}}^\perp}^2$ is sub-Weibull and concentrates around its mean $d - 1$. We replace the norm with $d$ in the population dynamics and absorb what remains into the sub-Weibull noise term (see Section B.4 for details on handling the noise). Hence, by a Taylor expansion,

$$\psi_\eta(y, z) = y\sigma'(z) + \sum_{k=2}^{r} \frac{(\eta d)^{k-1} (\sigma^{(k)}(z))(\sigma'(z))^{k-1}}{(k-1)!} y^k. \tag{4.3}$$

Note that Assumption 3.1 requires $\psi_\eta$ to be $O(1)$ and therefore $\eta \lesssim d^{-1}$. The quantity $\eta d$ controls the scaling of the higher order terms in the update. Indeed, we show in Appendix C.2 that

$$\mu_i(\eta) \asymp \sum_{k=1}^{r} \frac{(\eta d)^{k-1}}{(k-1)!} u_{i-1}(\sigma^{(k)}(\sigma')^{k-1}) u_i(\sigma_*^k). \tag{4.4}$$

**Corollary 4.2.** *Suppose Assumptions 3.1 and 3.2 hold, $\eta \lesssim d^{-1}$, $\gamma \lesssim \max_{1 \le i \le r} (\eta d)^{i-1} d^{-(\frac{p_i}{2} \vee 1)}$, and $\boldsymbol{w}^{(0)} \in \mathbb{S}^{d-1}$ such that $\langle \boldsymbol{\theta}_*, \boldsymbol{w}^{(0)} \rangle \asymp d^{-1/2}$. Then, with high probability,*

$$T(\eta) = \min_{1 \le i \le r} \tilde{\Theta}\big( (\eta d)^{-2(i-1)} d^{(p_i - 1) \vee 1} \big). \tag{4.5}$$

*iterations of Algorithm 1 are necessary and sufficient to achieve weak recovery.*

For any two distinct $i, j$ with $\mu_i, \mu_j > 0$, $\eta$ induces the phase transition:

$$(\eta d)^{-2(i-1)} d^{(p_i - 1) \vee 1} \le (\eta d)^{-2(j-1)} d^{(p_j - 1) \vee 1} \iff \eta \le d^{\frac{[(p_j - 1) \vee 1] - [(p_i - 1) \vee 1]}{2(j-i)} - 1}. \tag{4.6}$$

In particular, suppose that $u_{p_* - 1}(\sigma^{(I)} (\sigma')^{I-1}) u_{p_*}(\sigma_*^I) > 0$ and $u_p(\sigma_*) u_p(\sigma) > 0$ hold, which can be achieved with $\Theta(1)$ probability by a randomized activation agnostic to $\sigma_*$ as in [LOSW24]. Taking $\eta \lesssim d^{-\frac{p+1}{2}}$ gives the sample complexity $T = \Theta(d^{(p-1) \vee 1})$, which matches the online SGD bound [BAGJ21]. On the other hand, when $r \ge I$, taking $\eta \gtrsim d^{-1}$ as in [LOSW24] matches their sample complexity bound $T = \tilde{\Theta}(d)$. Intermediate values of $\eta$ interpolate between these two regimes. Appendix D details an experiment with batch reuse SGD exhibiting this phase transition.

### 4.3  Alternating SGD

Next, we consider a novel and simple variant of SGD that introduces a non-correlational update without changing the correlational loss. Algorithm 2, which we call *alternating SGD*, employs a two-step process to update $\boldsymbol{W}$. First, it computes a gradient update for $\boldsymbol{a}$ with learning rate $\eta$. Then, it uses the updated value $\tilde{a}$ in a gradient update on $\boldsymbol{W}$ with learning rate $\gamma$. We apply the same projected gradient and normalization to $\boldsymbol{w}$ as before. Crucially, the same sample $(\boldsymbol{x}, y)$ is used in these two updates. This produces a similar effect to batch reuse SGD without the need to apply consecutive gradient updates to $\boldsymbol{W}$. Specifically, the update to $\boldsymbol{W}$ in alternating SGD contains the label transformation $y \mapsto y^2$. This leads to a reduction in sample complexity if $p_2 := \mathbb{E}(\sigma_*^2) < p$ and the second layer learning rate $\eta$ is sufficiently large.

Algorithm 2 is related to studies on a two-timescales optimization dynamics of two-layer networks [BMZ24, MB23, BBPV23, WMHC24, BPV25], where training the second layer at a faster timescale simplifies the analysis. However, unlike these works, we perform sequential gradient updates on the two layers, and use the different learning rate scales to obtain a better sample complexity. We believe that the study of this algorithm is useful from a theoretical perspective and view it as proof of concept to demonstrate what mechanisms might be at play in neural network training so that they achieve near-optimal sample complexity.

---

**Algorithm 2:** Alternating SGD

**Input:** Learning rates $\eta, \gamma > 0$, sample size $T$
**Initialize** $\boldsymbol{w}^{(0)} \sim \text{Unif}(\mathbb{S}^{d-1})$, $a = 1$
**for** $t = 0$ *to* $t = T - 1$ **do**
  Draw i.i.d. sample $(\boldsymbol{x}, \boldsymbol{y})$
  Update $\tilde{a}^{(t+1)} \leftarrow a + \eta y \sigma(\langle \boldsymbol{x}, \boldsymbol{w}^{(t)} \rangle)$
  Update $\boldsymbol{w}^{(t+1)} \leftarrow \boldsymbol{w}^{(t)} + \gamma y \tilde{a}^{(t+1)} \sigma'(\langle \boldsymbol{x}, \boldsymbol{w}^{(t)} \rangle) \boldsymbol{P}_{\boldsymbol{w}^{(t)}}^\perp \boldsymbol{x}$
  Normalize $\boldsymbol{w}^{(t+1)} \leftarrow \boldsymbol{w}^{(t+1)} / \|\boldsymbol{w}^{(t+1)}\|$
**end**
**Output** $\boldsymbol{w}^{(T)}$

---

Given the second-layer gradient update $\tilde{a}^{(t+1)} = a + \eta y \sigma(\langle \boldsymbol{x}, \boldsymbol{w}^{(t)} \rangle)$, the update for $\boldsymbol{w}$ is

$$\psi(y, z) = ya\sigma'(z) + \eta y^2 \sigma(z) \sigma'(z). \tag{4.7}$$

Our calculation in Appendix C.3 yields the coefficients

$$\mu_i(\eta) = aiu_i(\sigma_*) u_i(\sigma) + \eta u_{i-1}(\sigma \sigma') u_i(\sigma_*^2). \tag{4.8}$$

The first term is identical to what emerges from the vanilla online SGD update (4.1). The second term arises from the non-correlational update. Note that if $\eta$ is chosen too small, the latter term may not dominate even when $p_2 < p$. The following makes this intuition rigorous.

**Corollary 4.3.** *Assume $\mu_p, \mu_{p_2} > 0$, $\eta \lesssim 1$, $\gamma \asymp \max\{d^{-(\frac{p}{2} \vee 1)}, \eta d^{-(\frac{p_2}{2} \vee 1)}\}$, and $\boldsymbol{w}^{(0)} \in \mathbb{S}^{d-1}$ such that $\langle \boldsymbol{\theta}_*, \boldsymbol{w}^{(0)} \rangle \asymp d^{-1/2}$. Then, with high probability,*

$$T(\eta) = \tilde{\Theta}\big(d^{(p-1)\vee 1}\big) \wedge \tilde{\Theta}\big(\eta^{-2}d^{(p_2-1)\vee 1}\big). \tag{4.9}$$

*iterations of Algorithm 2 are necessary and sufficient to achieve weak recovery.*

The assumption $\mu_p, \mu_{p_2} > 0$ is derived from our more general Assumption 3.2 and holds with $\Theta(1)$ probability if $\sigma$ follows the randomized construction in [LOSW24, Appendix B.1]. The sample complexity result implies a phase transition between the regime where the correlational term dominates and one where the non-correlational term dominates, occurring at (when $p \geq 2$)

$$d^{p-1} \asymp \eta^{-2}d^{(p_2-1)\vee 1} \iff \eta \asymp d^{-\frac{1}{2}[(p-p_2)\vee(p-2)]}. \tag{4.10}$$

In other words, alternating SGD improves over online SGD if squaring the target reduces its information exponent and $\eta$ is of strictly larger order than the threshold above. For example, if $\sigma_* = \mathsf{He}_3$, then $p = 3$ and $p_2 = 2$. The phase transition occurs at $\eta \asymp d^{-1/2}$. At or below this threshold, alternating SGD has quadratic complexity, while $\eta \gtrsim 1/\operatorname{polylog} d$ gives linear complexity (up to polylogarithmic factors). Meanwhile, intermediate values of $\eta$ interpolate between these two regimes.

Figure 1 illustrates a simulation for the above toy example. As predicted by Corollary 4.3, we observe two distinct phases. In the first phase, the number of samples required to achieve small test error is constant in $\eta$ until a critical threshold is reached, at which point the second phase begins and the test error decreases at a $\tilde{\Theta}(1/\eta^2)$ rate.

**Extension to Deeper Networks.** It is possible to further improve the sample complexity for alternating SGD in the case $p_2 > 2$ by employing a deeper (but sparse) neural network. The natural generalization of the algorithm is to take a gradient step on each layer while keeping the remaining layers frozen, starting from the outermost layer. In Appendix C.4, we show that, under assumptions on the Hermite coefficients of compositions of activation functions[5], this fits into our framework with $\mu_i(\eta) \asymp \sum_{j=1}^{D} \eta^{j-1} u_i(\sigma_*^j)$ and

$$T = \max_{1 \leq i \leq D} \tilde{\Theta}\big(\eta^{-2(i-1)}d^{(p_i-1)\vee 1}\big), \tag{4.11}$$

where $D$ is the number of layers. The number of phase transitions in $\eta$ is one less than the number of distinct $p_i = \mathrm{IE}(\sigma_*^i)$, $i \in [D]$. Moreover, if $\eta \gtrsim 1/\operatorname{polylog} d$ and $D \geq I$, where $I$ is as in Lemma 2.3, the sample complexity is $\tilde{\Theta}(d)$. As we will see in the next subsection, depth $D$ plays an analogous role to the degree of $\sigma$ in batch reuse SGD.

## 5 Proof Sketch

The complete proof of Theorem 3.3 in Appendix B is inspired by and builds on previous analyses of online algorithms [BAGJ21, LOSW24]. We outline the main steps here. We focus on the sample complexity upper bound; the proof of the matching lower bound is similar.

First, we derive the expected dynamics for a single update (3.1). The key quantity in this step is the alignment between the one-step update function $\boldsymbol{g}^{(t)} = \psi_\eta(y, \langle \boldsymbol{x}^{(t)}, \boldsymbol{w}^{(t)} \rangle) \boldsymbol{P}_{\boldsymbol{w}^{(t)}}^{\perp} \boldsymbol{x}^{(t)}$ and the target direction $\boldsymbol{\theta}_*$, which has expectation

$$\mathbb{E}[\langle \boldsymbol{\theta}_*, \boldsymbol{g}^{(t)} \rangle] = \mathbb{E}_{\boldsymbol{x}}\big[\psi_\eta(y, \langle \boldsymbol{x}, \boldsymbol{w}^{(t)} \rangle) \langle \boldsymbol{P}_{\boldsymbol{w}^{(t)}}^{\perp} \boldsymbol{x}, \boldsymbol{\theta}_* \rangle\big] = \sum_{i=1}^{r} i! \mu_i \langle \boldsymbol{\theta}_*, \boldsymbol{w}^{(t)} \rangle^{i-1} \big(1 - \langle \boldsymbol{\theta}_*, \boldsymbol{w}^{(t)} \rangle^2\big), \tag{5.1}$$

where we have hidden the conditioning on previous iterations for convenience. The last equality is obtained by taking the Hermite expansion of $\psi_\eta$ in each of its arguments (hence the appearance of the $\mu_i$ defined in (3.2)) and applying Stein's Lemma. Now, what distinguishes our framework is that we do not assume that the nonzero $\mu_i$ are $\tilde{\Theta}(1)$. In the analysis of online SGD in [BAGJ21], $\mu_p$ is the

---

[5]These assumptions are difficult to verify in general. However, in the same appendix, we show that they hold for a three-layer network with $\sigma(z) = z^2$ and target satisfying $u_2(\sigma_*^3) > 0$, in which case weak recovery occurs in $\tilde{\Theta}(d)$ iterations when $\eta \asymp 1$.

first non-zero coefficient, and thus the $p$th term in the above sum dominates. Similarly, nonzero $\mu_{p_*}$ yields the dominant term in the analysis of batch reuse SGD. On the other hand, we allow for the situation where the first nonzero $\mu_i$ is of strictly smaller order than some $\mu_j$ with $j > i$ due to small $\eta$.

The resulting one-step dynamics are

$$\langle \boldsymbol{\theta}_*, \boldsymbol{w}^{(t+1)} \rangle \geq \langle \boldsymbol{\theta}_*, \boldsymbol{w}^{(t)} \rangle + \gamma \tilde{C}_1 \sum_{i=1}^{r} \mu_i \langle \boldsymbol{\theta}_*, \boldsymbol{w}^{(t)} \rangle^{i-1} + \gamma \nu^{(t)} - \gamma^2 d \langle \boldsymbol{\theta}_*, \boldsymbol{w}^{(t)} \rangle, \qquad (5.2)$$

where $\tilde{C}_1 > 0$ is a constant, $\nu^{(t)}$ is a sub-Weibull random variable independent from prior iterations that results from the label noise and the randomness in the input $\boldsymbol{x}$, and the final term is a bound on the impact of the normalization step in (3.1). The latter term can be controlled and absorbed into the expected update term when $\gamma \lesssim \max_{1 \leq i \leq r} \mu_i d^{-(\frac{1}{2} \vee 1)}$.

Next, we unravel the recurrence to obtain

$$\langle \boldsymbol{\theta}_*, \boldsymbol{w}^{(t)} \rangle \geq \langle \boldsymbol{\theta}_*, \boldsymbol{w}^{(0)} \rangle + \gamma C_1 \sum_{i=1}^{r} \sum_{s=0}^{t-1} \mu_i \langle \boldsymbol{\theta}_*, \boldsymbol{w}^{(s)} \rangle^{i-1} - \gamma \left| \sum_{s=0}^{t-1} \nu^{(s)} \right| \qquad (5.3)$$

and control the last term with martingale and sub-Weibull concentration bounds. We identify which of the terms in the expected update dominate by bounding the sequence

$$\alpha^{(t)} = d^{-1/2} + \gamma \sum_{i=1}^{r} \mu_i (\alpha^{(s)})^{i-1}, \quad \alpha^{(0)} = d^{-1/2}, \qquad (5.4)$$

leveraging Grönwall's Inequality (Lemma A.3) for the $i = 2$ term and the Bihari-LaSalle Inequality (Lemma A.4) for the $i \geq 3$ terms. Setting this equal to $c \gtrsim 1/\operatorname{polylog} d$ yields the sample complexity.

## 6   Conclusion

This work demonstrates that the learning rate is a fundamental factor in determining the sample complexity of gradient-based algorithms for learning single-index models with neural networks. We show that algorithms that employ a combination of correlational and non-correlational update terms (with separate learning rates) exhibit a phase transition between distinct sample complexity regimes as a function of the scaling of the non-correlational term. In both our novel alternating SGD and the batch reuse algorithm of [DTA+24, LOSW24, ADK+24b], this scaling manifests as a learning rate $\eta$ that appears in the first of a two-step update. If $\eta$ is chosen too small, then the sample complexity is no better than the $n = \tilde{\Theta}(d^{(p-1)\vee 1})$ bound for online SGD. On the other hand, when $\eta$ increases beyond the phase transition threshold, it interpolates between this information exponent regime and a generative exponent regime where the rate becomes $n = \tilde{\Theta}(d^{(p_*-1)\vee 1})$.

In addition to illustrating of the phase transition, our novel alternating SGD algorithm presents an alternative to batch reuse and changing the loss for improving upon the CSQ sample complexity. Interestingly, it admits a natural generalization to neural networks with more than 2 layers that enlarge the class of target polynomials for which weak recovery can be achieved with linear (up to poly-logarithmic factors) sample complexity. These findings open the door to investigating theoretically tractable settings where the relationship between depth and sample complexity can be precisely quantified.

Other natural directions for future work include an extension of our framework to multi-index models [AAM23], more general input distributions [JKMS25], non-polynomial activation functions, and non-constant learning rates. Given the generality of our update oracle $\psi_\eta$, we also expect that our framework can be to adapted to capture landscape smoothing [DNGL23] — as well as similar algorithms that aggregate gradients evaluated at perturbed weights [CWL+25] — with the associated hyperparameter $\lambda$ being cast as our $\eta$.

## Acknowledgments and Disclosure of Funding

The authors thank Denny Wu and Jivan Waber for helpful discussions. Resources used in preparing this research were provided, in part, by the Province of Ontario, the Government of Canada through

CIFAR, and companies sponsoring the Vector Institute. KCT was supported by NSERC through the PGS-D program. MAE was partially supported by the NSERC Grant [2019-06167], the CIFAR AI Chairs program, the CIFAR Catalyst grant, and the Ontario Early Researcher Award.

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

# A Technical Background

We synthesize and reference the key technical background that forms the backbone of our proofs that appear in Appendix B.

## A.1 Hermite Polynomials

**Definition A.1** (Probabilist's Hermite Polynomials [O'D21, Definition 11.29]). *The probabilist's Hermite polynomials* $\mathsf{He}_j : \mathbb{R} \to \mathbb{R}$, $j \in \mathbb{N}_0$, *are defined as*

$$\mathsf{He}_j(z) = (-1)^j e^{\frac{z^2}{2}} \frac{d^j}{dz^j} \left[ e^{-\frac{z^2}{2}} \right]. \tag{A.1}$$

For example, the first four probabilist's Hermite polynomials are $\mathsf{He}_0(z) = 1$, $\mathsf{He}_1(z) = z$, $\mathsf{He}_2(z) = z^2 - 1$, and $\mathsf{He}_3(z) = z^3 - 3z$.

It is well-known that $\{\mathsf{He}_j\}_{j=0}^\infty$ form an orthogonal basis for $L^2(\mathcal{N}(0,1))$. In particular,

$$\mathbb{E}_{z \sim \mathcal{N}(0,1)}[\mathsf{He}_i(z)\mathsf{He}_j(z)] = j! \delta_{i=j}. \tag{A.2}$$

Since our proofs rely on the analysis of Hermite polynomials applied to inner products, the following consequence of orthonormality is particularly useful.

**Lemma A.2** ([O'D21, Proposition 11.31]). *Suppose that* $z, z' \sim \mathcal{N}(0,1)$ *such that* $\mathrm{Cov}(z, z') = \rho$. *Then,*

$$\mathbb{E}_{z,z'} \left[ \mathsf{He}_i(z)\mathsf{He}_j(z') \right] = j! \rho^j \delta_{i=j}. \tag{A.3}$$

In particular, for $\boldsymbol{x} \sim \mathcal{N}(0, \boldsymbol{I}_d)$, $\boldsymbol{w} \in \mathbb{R}^d$, $\boldsymbol{\theta}_* \in \mathbb{R}^d$, we have

$$\mathbb{E}_{\boldsymbol{x}} \left[ \mathsf{He}_i(\langle \boldsymbol{x}, \boldsymbol{w} \rangle)\mathsf{He}_j(\langle \boldsymbol{x}, \boldsymbol{\theta}_* \rangle) \right] = j! \langle \boldsymbol{w}, \boldsymbol{\theta}_* \rangle^j \delta_{i=j}. \tag{A.4}$$

## A.2 Discrete-Time Dynamical Systems

**Lemma A.3** (Discrete Grönwall Inequality [Cla87]). *Let* $\{m_t\}_{t=0}^\infty$ *be a sequence such that* $m_0 = a$ *and* $m_t \le a + c \sum_{j=0}^{t-1} m_j$ *for all* $t \ge 1$, *where* $a, c > 0$. *Then, for all* $t \ge 0$,

$$m_t \le a(1+c)^t \le a e^{ct}. \tag{A.5}$$

*Moreover, if instead* $m_t \ge a + c \sum_{j=0}^{t-1} m_j$ *for all* $t \ge 1$, *then* $m_t \ge a(1+c)^t$ *for all* $t \ge 0$.

*Proof.* The result easily follows by induction. The statement is trivial for $t = 0$. Suppose now that it holds for some $t \ge 0$. Then,

$$m_{t+1} \le a + c \sum_{j=0}^t m_j \le a + c \sum_{j=0}^t a(1+c)^j = a + ca \left( \frac{(1+c)^{t+1} - 1}{(1+c) - 1} \right) = a(1+c)^{t+1}. \tag{A.6}$$

The same argument can be used for the reversed inequality. □

**Lemma A.4** (Discrete Bihari-LaSalle Inequality [BAGJ21, Appendix C]; [LOSW24, Lemma 18]). *Let* $\{m_t\}_{t=0}^\infty$ *be a sequence such that* $m_0 = a$ *and* $m_t \le a + c \sum_{j=0}^{t-1} m_j^{k-1}$ *for all* $t \ge 1$, *where* $a, c > 0$, *and* $k \ge 3$. *Then,*

$$m_t \le \frac{a}{(1 - (k-2)ca^{k-2}t)^{\frac{1}{k-2}}}, \quad \forall 0 \le t < \frac{1}{c(k-2)a^{k-2}}. \tag{A.7}$$

*Moreover, if instead* $m_t \ge a + c \sum_{j=0}^{t-1} m_j^{k-1}$, *then*

$$m_t \ge \frac{a}{(1 - \frac{c}{2}a^{k-2}t)^{\frac{1}{k-2}}}, \quad \forall 0 \le t < \frac{1}{c(k-2)}(a^{-(k-2)} - c). \tag{A.8}$$

*Proof.* Let $\{a_t\}_{t=0}^{\infty}$ be such that $a_0 = a$ and $a_t = a + c\sum_{j=0}^{t-1} a_j^{k-1}$.

**Upper Bound.** Define $\{b_t\}_{t=0}^{\infty}$ by $b_0 = a$ and $b_t = a + \sum_{j=0}^{t-1} c(m_j)^{k-1}$. Then, $m_t \leq b_t$ by definition. We prove that $b_t \leq a_t$ by induction. Clearly, $b_0 = a_0$. Now,

$$b_{t+1} = a + \sum_{j=0}^{t} c(m_j)^{k-1} = b_t + c(m_t)^{k-1} \leq b_t + c(b_t)^{k-1} \leq a_t + c(a_t)^{k-1} = a_{t+1}, \quad \text{(A.9)}$$

where the last inequality follows from the induction hypothesis. Hence, $m_t \leq b_t \leq a_t$ for all $t \geq 0$.

Notice for all $t \geq 1$ that

$$c = \frac{a_{t+1} - a_t}{a_t^{k-1}} = \int_{a_t}^{a_{t+1}} \frac{1}{a_t^{k-1}} \, dx \geq \int_{a_t}^{a_{t+1}} \frac{1}{x^{k-1}} \, dx = \frac{1}{k-2}\left(\frac{1}{a_t^{k-2}} - \frac{1}{a_{t+1}^{k-2}}\right). \quad \text{(A.10)}$$

Rearranging the above, we have

$$a_{t+1}^{-(k-2)} \geq a_t^{-(k-2)} - c(k-2). \quad \text{(A.11)}$$

Unrolling the recurrence, we obtain

$$a_t^{-(k-2)} \geq a_0^{-(k-2)} - c(k-2)t. \quad \text{(A.12)}$$

So long as $a^{-(k-2)} - c(k-2)t > 0$, we can rearrange to obtain the desired upper bound

$$a_t \leq \frac{1}{\left(a_0^{-(k-2)} - c(k-2)t\right)^{\frac{1}{k-2}}} = \frac{a}{\left(1 - (k-2)ca^{k-2}t\right)^{\frac{1}{k-2}}}. \quad \text{(A.13)}$$

The condition $a^{(k-2)} - c(k-2)t > 0$ holds so long as

$$t < \frac{1}{c(k-2)a^{k-2}} = \Theta(a^{-(k-2)}), \quad \text{(A.14)}$$

matching the condition in (A.7).

**Lower Bound.** A similar induction argument to the one in the upper bound proof shows that $m_t \geq a_t$ for all $t \geq 0$.

For each $t \geq 0$, let $b_t = a_t^{-(k-2)}$. Rewriting a step of the recurrence as

$$a_{t+1} = a_t\left(1 + \frac{c}{a_t^{-(k-2)}}\right) \quad \text{(A.15)}$$

allows us to write a recurrence for $\{b_t\}_{t=0}^{\infty}$:

$$b_{t+1} = b_t\left(1 + \frac{c}{b_t}\right)^{-(k-2)} \leq b_t\left(\frac{1}{1 + \frac{c}{b_t}}\right) = \frac{b_t}{\frac{b_t+c}{b_t}} = \frac{b_t^2}{b_t + c} = b_t - \frac{cb_t}{b_t + c}. \quad \text{(A.16)}$$

Now, so long as $b_t \geq c$, we have $b_{t+1} \leq b_t - \frac{c}{2}$. Unrolling the recurrence and rewriting in terms of the $a_t$ gives

$$\begin{aligned} b_t &\leq b_0 - \frac{c}{2}t \\ \Rightarrow a_t^{-(k-2)} &\leq a_0^{-(k-2)} - \frac{c}{2}t \\ \Rightarrow a_t &\geq \frac{1}{\left(a_0^{-(k-2)} - \frac{c}{2}t\right)^{\frac{1}{k-2}}} = \frac{a}{\left(1 - \frac{c}{2}a^{k-2}t\right)^{\frac{1}{k-2}}}. \end{aligned} \quad \text{(A.17)}$$

It remains to characterize $t$ for which $b_t \geq c$ holds. Recall from (A.12) that $b_t \geq b_0 - c(k-2)t$ for all $t \geq 0$. Notice

$$b_0 - c(k-2)t \geq c \iff t \leq \frac{b_0 - c}{c(k-2)}. \quad \text{(A.18)}$$

which matches the condition on $t$ in (A.8). $\qquad \square$

# B    Proof of Main Result

We follow a very similar line of reasoning to the proofs of other sample complexity bounds involving the information and generative exponent in the literature, e.g., [BAGJ21, LOSW24]. Given a sample $(\boldsymbol{x}, y)$ from the target single-index model (2.1), recall from Section 3 that the update equation for $\boldsymbol{w}$ is

$$\boldsymbol{w}^{(t+1)} = \frac{\boldsymbol{w}^{(t)} + \gamma\psi_\eta(y, \langle \boldsymbol{x}, \boldsymbol{w}^{(t)}\rangle)\boldsymbol{P}^\perp_{\boldsymbol{w}^{(t)}}\boldsymbol{x}}{||\boldsymbol{w}^{(t)} + \gamma\psi_\eta(y, \langle \boldsymbol{x}, \boldsymbol{w}^{(t)}\rangle)\boldsymbol{P}^\perp_{\boldsymbol{w}^{(t)}}\boldsymbol{x}||}, \tag{B.1}$$

where $\boldsymbol{P}^\perp_{\boldsymbol{w}} = \boldsymbol{I}_d - \boldsymbol{w}\boldsymbol{w}^\top$. Throughout this section, we adopt the notation $\kappa^{(t)} = \langle \boldsymbol{\theta}_*, \boldsymbol{w}^{(t)}\rangle$ and $\boldsymbol{g}(\boldsymbol{w}; \boldsymbol{x}, y) = \psi_\eta(y, \langle \boldsymbol{x}, \boldsymbol{w}\rangle)\boldsymbol{P}^\perp_{\boldsymbol{w}}\boldsymbol{x}$. We are interested in the dynamics of the alignment with the ground truth

$$\kappa^{(t+1)} = \frac{\kappa^{(t)} + \gamma\langle \boldsymbol{\theta}_*, \boldsymbol{g}^{(t)}\rangle}{||\boldsymbol{w}^{(t)} + \gamma\boldsymbol{g}^{(t)}||}. \tag{B.2}$$

In Section B.1, using standard Gaussian tail bounds and tools from high-dimensional probability, we characterize the concentration of the initial alignment $\kappa^{(0)}$ about $d^{-\frac{1}{2}}$. Next, in Section B.2, we describe the "slowdown" in the alignment dynamics due to normalization. In Section B.3, we lower bound the expected update after one step. In Section B.4, we expand the expected dynamics over $t$ steps and employ a standard martingale bound to control the noise, leading to a high probability upper bound on sample complexity when the initial alignment is of order $d^{-\frac{1}{2}}$. This is complemented by a matching lower bound proven in the same way in Section B.5. The upper and lower bounds immediately imply Theorem 3.3, our main result. Subsequently, we discuss how weak recovery leads to strong recovery (Section B.6) and approximation of the target to arbitrary accuracy (Section B.7). This will elucidate the fact that achieving weak recovery is the sample complexity bottleneck for any generic online algorithm satisfying our formalism in Section 3.

## B.1    Initial Alignment

We follow [LOSW24] in showing a high-probability lower bound for the alignment between a hidden neuron's weight vector $\boldsymbol{w}$ and the ground truth direction $\boldsymbol{\theta}_*$. We make a small modification to remove the dependence on step size from the bound.

**Lemma B.1.** *Let $\boldsymbol{w}^{(0)} \sim \mathrm{Unif}(\mathbb{S}^{d-1})$. Then, $\mathbb{P}(\kappa^{(0)} \geq C_0 d^{-1/2}) = \Omega(1)$ for any constant $C_0 > 0$. Moreover, for any $\delta' > 0$ there exists $\tilde{C}_0 \geq r$ such that $\mathbb{P}(\kappa^{(0)} \geq \tilde{C}_0 d^{-1/2}) \leq \delta'$.*

*Proof.* We may write

$$\kappa^{(0)} = \langle \boldsymbol{\theta}_*, \boldsymbol{w}^{(0)}\rangle \overset{d}{=} \frac{\langle \boldsymbol{e}_1, \boldsymbol{g}\rangle}{||\boldsymbol{g}||}, \tag{B.3}$$

where $\boldsymbol{e}_1 \in \mathbb{R}^d$ is the first standard basis vector and $\boldsymbol{g} \sim \mathcal{N}(\boldsymbol{0}, \boldsymbol{I}_d)$.

To proceed, as in [LOSW24], we require the following lemma.

**Lemma B.2** ([CCM11, Theorem 2]). *For any $\beta > 1$ and $s \in \mathbb{R}$, we have*

$$\frac{\sqrt{2e(\beta-1)}}{2\beta\sqrt{\pi}}e^{-\frac{\beta s^2}{2}} \leq \int_s^\infty \frac{1}{\sqrt{2\pi}}e^{-\frac{t^2}{2}}\,dt. \tag{B.4}$$

Then,

$$\begin{aligned}
\mathbb{P}(\kappa^{(0)} \geq C_0 d^{-1/2}) &\geq \mathbb{P}(\langle \boldsymbol{e}_1, \boldsymbol{g}\rangle \geq 2C_0 \wedge ||\boldsymbol{g}|| \leq C_0 d^{1/2}) \\
&\geq \mathbb{P}(\langle \boldsymbol{e}_1, \boldsymbol{g}\rangle \geq 2C_0) - \mathbb{P}(||\boldsymbol{g}|| \geq C_0 d^{-1/2}) \\
&\geq \frac{\sqrt{2e(\beta-1)}}{2\beta\sqrt{\pi}}e^{-2C_0^2\beta} - e^{-\Omega(d)},
\end{aligned} \tag{B.5}$$

where the second term follows from Gaussian concentration of the norm. Taking $\beta = 2$, we see that the above is $\Theta(1)$.

We can derive a high probability bound using Lipschitz concentration [Ver18, Theorem 5.1.4] to obtain

$$\mathbb{P}\big(|\kappa^{(0)}| \geq \tilde{C}_0 d^{-1/2}\big) \leq 2\exp(-\tilde{c}\tilde{C}_0^2). \tag{B.6}$$

for some $\tilde{c} > 0$. Arguing by symmetry and taking $\tilde{C}_0$ sufficiently large gives the second part of the result. $\qquad\square$

## B.2 Normalization Error

**Lemma B.3.** *Suppose $\kappa^{(t)} \geq 0$. The update* (B.2) *satisfies the lower bound*

$$\kappa^{(t+1)} \geq \kappa^{(t)} + \gamma\langle\boldsymbol{\theta}_*, \boldsymbol{g}^{(t)}\rangle - \gamma^2\kappa^{(t)}||\boldsymbol{g}^{(t)}||^2 - \gamma^3|\langle\boldsymbol{\theta}_*, \boldsymbol{g}^{(t)}\rangle|\,||\boldsymbol{g}^{(t)}||^2. \tag{B.7}$$

*Proof.* When $\kappa^{(t)} + \gamma\langle\boldsymbol{\theta}_*, \boldsymbol{g}^{(t)}\rangle \geq 0$, we have

$$
\begin{aligned}
\kappa^{(t+1)} &= \frac{\kappa^{(t)} + \gamma\langle\boldsymbol{\theta}_*, \boldsymbol{g}^{(t)}\rangle}{||\boldsymbol{w}^{(t)} + \gamma\boldsymbol{g}^{(t)}||} \\
&= \frac{\kappa^{(t)} + \gamma\langle\boldsymbol{\theta}_*, \boldsymbol{g}^{(t)}\rangle}{\sqrt{1 + \gamma^2||\boldsymbol{g}^{(t)}||^2}} \\
&\geq (\kappa^{(t)} + \gamma\langle\boldsymbol{\theta}_*, \boldsymbol{g}^{(t)}\rangle)(1 - \gamma^2||\boldsymbol{g}^{(t)}||^2) \\
&\geq \kappa^{(t)} + \gamma\langle\boldsymbol{\theta}_*, \boldsymbol{g}^{(t)}\rangle - \kappa^{(t)}\gamma^2||\boldsymbol{g}^{(t)}||_2^2 - \gamma^3|\langle\boldsymbol{\theta}_*, \boldsymbol{g}^{(t)}\rangle|\,||\boldsymbol{g}^{(t)}||_2^2.
\end{aligned} \tag{B.8}
$$

The second line follows from the facts $\langle\boldsymbol{w}^{(t)}, \boldsymbol{g}^{(t)}\rangle = 0$ (due to $\boldsymbol{P}_{\boldsymbol{w}}^{\perp}$) and $\boldsymbol{w}^{(t)} \in \mathbb{S}^{d-1}$. The third line is trivial if $\gamma^2||\boldsymbol{g}^{(t)}||^2 \geq 1$. Otherwise, observe that when $\gamma^2||\boldsymbol{g}^{(t)}||^2 < 1$,

$$
\begin{aligned}
1 - \gamma^2||\boldsymbol{g}^{(t)}||^2 &\leq \frac{1}{\sqrt{1 + \gamma^2||\boldsymbol{g}^{(t)}||^2}} \\
\iff \big(1 - \gamma^2||\boldsymbol{g}^{(t)}||^2\big)^2\big(1 + \gamma^2||\boldsymbol{g}^{(t)}||^2\big) &\leq 1 \\
\iff \big(1 - \gamma^4||\boldsymbol{g}^{(t)}||^2\big)\big(1 - \gamma^2||\boldsymbol{g}^{(t)}||^2\big) &\leq 1,
\end{aligned} \tag{B.9}
$$

where the last line clearly holds. Now, when $\kappa^{(t)} + \gamma\langle\boldsymbol{\theta}_*, \boldsymbol{g}^{(t)}\rangle < 0$, the same lower bound can be shown via

$$
\begin{aligned}
\kappa^{(t)} + \gamma\langle\boldsymbol{\theta}_*, \boldsymbol{g}^{(t)}\rangle - \kappa^{(t)}\gamma^2||\boldsymbol{g}^{(t)}||_2^2 - \gamma^3|\langle\boldsymbol{\theta}_*, \boldsymbol{g}^{(t)}\rangle|\,||\boldsymbol{g}^{(t)}||_2^2 &\leq \kappa^{(t)} + \gamma\langle\boldsymbol{\theta}_*, \boldsymbol{g}^{(t)}\rangle \\
&\leq \frac{\kappa^{(t)} + \gamma\langle\boldsymbol{\theta}_*, \boldsymbol{g}^{(t)}\rangle}{(1 + \gamma^2||\boldsymbol{g}^{(t)}||^2)^{1/2}}.
\end{aligned} \tag{B.10}
$$

$\square$

## B.3 One-Step Population Dynamics

We extend the definition of the coefficients $\mu_i$ from (3.2) to handle label noise. Define

$$\mu_i := \underset{(a,b)\sim\mathcal{N}(\boldsymbol{0}, \boldsymbol{I}_2)}{\mathbb{E}} \mathbb{E}_\zeta\left[\psi_\eta(\sigma_*(a) + \zeta, b)\mathsf{He}_i(a)\mathsf{He}_{i-1}(b)\right], \quad i \in [r]. \tag{B.11}$$

**Lemma B.4.** *Assume that for some $t \geq 0$ we have $d^{-\frac{1}{2}} \leq \kappa^{(t)} \lesssim \frac{1}{\text{polylog}\,d}$. Moreover, suppose that Assumption 3.2 holds. Then, there exists $C > 0$ such that taking $\gamma \leq C\max_{1\leq i\leq r} \mu_i d^{-(\frac{i}{2}\vee 1)}$ yields the following lower bound for the one-step dynamics of the alignment $\kappa^{(t)} := \langle\boldsymbol{\theta}_*, \boldsymbol{w}^{(t)}\rangle$:*

$$\kappa^{(t+1)} \geq \kappa^{(t)} + \gamma C_1\sum_{i=1}^{r}\mu_i(\kappa^{(t)})^{i-1}(1 - (\kappa^{(t)})^2) + \gamma\nu^{(t)}, \tag{B.12}$$

*where $\nu^{(t)}$ is a mean-zero sub-Weibull random variable with $\Theta(1)$ tail parameter and $C_1 > 0$ is a constant.*

*Proof.* Omitting the superscript $t$, the expected update to the alignment with the ground truth $\boldsymbol{\theta}_*$ (given the previous iterate) is

$$
\begin{aligned}
\mathbb{E}[\langle \boldsymbol{\theta}_*, \boldsymbol{g}\rangle] &= \boldsymbol{\theta}_*^\top \boldsymbol{P}_{\boldsymbol{w}}^\perp \mathbb{E}_{\boldsymbol{x},y}\big[\psi_\eta\big(y, \langle \boldsymbol{x}, \boldsymbol{w}\rangle\big)\boldsymbol{x}\big] \\
&= \boldsymbol{\theta}_*^\top \boldsymbol{P}_{\boldsymbol{w}}^\perp \mathbb{E}_{\boldsymbol{x},\zeta}\Bigg[\sum_{j=0}^r \sum_{i=0}^\infty \mathbb{E}_{a,b}\big[\psi_\eta(\sigma_*(a) + \zeta, b)\mathsf{He}_i(a)\mathsf{He}_j(b)\big]\mathsf{He}_i(\langle \boldsymbol{x}, \boldsymbol{\theta}_*\rangle)\mathsf{He}_j(\langle \boldsymbol{x}, \boldsymbol{w}\rangle)\boldsymbol{x}\Bigg] \\
&= \sum_{i=1}^\infty \sum_{j=0}^r \mathbb{E}_{\boldsymbol{x},\zeta}\Big[\mu_i i \mathsf{He}_{i-1}(\langle \boldsymbol{x}, \boldsymbol{w}\rangle)\mathsf{He}_j(\langle \boldsymbol{x}, \boldsymbol{\theta}_*\rangle)\Big]\langle \boldsymbol{\theta}_*, \boldsymbol{P}_{\boldsymbol{w}}^\perp \boldsymbol{\theta}_*\rangle \\
&= \sum_{i=1}^r i! \mu_i \langle \boldsymbol{\theta}_*, \boldsymbol{w}\rangle^{i-1}\big(1 - \langle \boldsymbol{\theta}_*, \boldsymbol{w}\rangle^2\big),
\end{aligned}
\tag{B.13}
$$

where the third line uses Stein's Lemma and the fact $\boldsymbol{P}_{\boldsymbol{w}}^\perp \boldsymbol{w} = \boldsymbol{0}$. Thus, the size of the update will be dictated by the first index $i_*$ such that $|\mu_i|\langle \boldsymbol{\theta}_*, \boldsymbol{w}\rangle^{i-1}$ is largest. Moreover, the centred random variable $\langle \boldsymbol{\theta}_*, \boldsymbol{g}\rangle - \mathbb{E}[\langle \boldsymbol{\theta}_*, \boldsymbol{g}\rangle]$ is sub-Weibull with constant order tail parameter since Gaussian random variables are sub-Weibull and the latter class is closed under polynomial transformation. (See [VGNA20] for more details on sub-Weibull random variables).

We must also control the normalization error from Lemma B.3:

$$
\gamma^2 \kappa^{(t)} ||\boldsymbol{g}||^2 + \gamma^3 |\langle \boldsymbol{\theta}_*, \boldsymbol{g}\rangle| \, ||\boldsymbol{g}||^2.
\tag{B.14}
$$

Note that

$$
\begin{aligned}
||\boldsymbol{g}||^2 &= ||\boldsymbol{P}_{\boldsymbol{w}}^\perp \boldsymbol{x}||^2 \big|\psi_\eta(y, \langle \boldsymbol{x}, \boldsymbol{w}\rangle)\big|^2 \\
&= \Bigg|\sum_{j=0}^r \mathbb{E}_b[\psi_\eta(y, b)\mathsf{He}_j(b)]\mathsf{He}_j(\langle \boldsymbol{x}, \boldsymbol{w}\rangle)\Bigg|^2 (||\boldsymbol{x}||^2 - \langle \boldsymbol{x}, \boldsymbol{w}\rangle^2),
\end{aligned}
\tag{B.15}
$$

and therefore, since $||\boldsymbol{x}||$ and $\langle \boldsymbol{x}, \boldsymbol{w}\rangle$ are independent,

$$
\mathbb{E}\big[||\boldsymbol{g}||^2\big] = \mathbb{E}_{\boldsymbol{x}}\Bigg[\Bigg|\sum_{j=0}^r \mathbb{E}_b[\psi_\eta(y, b)\mathsf{He}_j(b)]\mathsf{He}_j(\langle \boldsymbol{x}, \boldsymbol{w}\rangle)\Bigg|(d - \langle \boldsymbol{x}, \boldsymbol{w}\rangle^2)\Bigg] \lesssim d.
\tag{B.16}
$$

By the same token, we use our derivation in (B.13) to argue $\mathbb{E}[|\langle \boldsymbol{\theta}_*, \boldsymbol{g}\rangle|\,||\boldsymbol{g}||^2] \lesssim d$. The error (B.15) is a sub-Weibull random variable with tail parameter proportional to $\gamma^2 d$.

By Lemma B.3, this implies that the one step dynamics take the form

$$
\kappa^{(t+1)} \geq \kappa^{(t)} + \gamma \mathbb{E}[\langle \boldsymbol{\theta}_*^{(t)}, \boldsymbol{g}^{(t)}\rangle] + \gamma \nu^{(t)} - C_2 \gamma^2 \kappa^{(t)}(d + \xi^{(t)}).
\tag{B.17}
$$

for some positive constant $C_2$ and sub-Weibull random variables (with constant order parameter) $\nu^{(t)}, \xi^{(t)}$ that are independent of previous iterations. Now, choosing $\gamma \leq C \max_{1 \leq i \leq r} \mu_i d^{-(\frac{i}{2} \vee 1)}$ and recalling $\kappa^{(t)} \geq d^{-1/2}$ leads to

$$
C_2 \gamma^2 \kappa^{(t)} d \leq C_2 C \gamma \kappa^{(t)} \max_{1 \leq i \leq r} \mu_i d^{-(\frac{i-2}{2} \vee 0)} \leq C_2 C \gamma \max_{1 \leq i \leq r} \mu_i (\kappa^{(t)})^{(i-1) \vee 1},
\tag{B.18}
$$

which can be made a sufficiently small constant multiple of $\gamma \max_{1 \leq i \leq r} \mu_i (\kappa^{(t)})^{i-1}$ with an appropriate choice of $C$. Hence, the expected one-step normalization error can be absorbed into the expected one-step population dynamics (B.13). Furthermore, since the constraint on $\gamma$ also implies $\gamma d \lesssim 1$, we may absorb the noise $\gamma^2 d\xi^{(t)}$ into the $\gamma \nu^{(t)}$ noise term.

This leaves us with the alignment dynamics

$$
\kappa^{(t+1)} \geq \kappa^{(t)} + \gamma C_1 \sum_{i=1}^r \mu_i (\kappa^{(t)})^{i-1}\big(1 - (\kappa^{(t)})^2\big) + \gamma \nu^{(t)}
\tag{B.19}
$$

for some constant $C_1 > 0$.

$\qquad\square$

## B.4 Sample Complexity Upper Bound

**Proposition B.5** (Generic Sample Complexity Upper Bound). *Suppose $\langle \boldsymbol{\theta}_*, \boldsymbol{w}^{(0)} \rangle \geq C_0 d^{-1/2}$ for some $C_0 > 0$. Then, there exists $C \gtrsim 1/\operatorname{polylog} d$ such that for any $\delta \in (0,1)$, setting $\gamma \leq C\delta \max_{1 \leq i \leq r} \mu_i d^{-(\frac{i}{2} \vee 1)}$ gives $\langle \boldsymbol{\theta}_*, \boldsymbol{w}^{(t)} \rangle \gtrsim 1/\operatorname{polylog} d$ within*

$$T(\eta) = \min_{\substack{1 \leq i \leq r \\ \mu_i > 0}} \tilde{\Theta}\big(\gamma^{-1}\big(\mu_i(\eta)\big)^{-1} d^{\frac{i-2}{2} \vee 0}\big) \tag{B.20}$$

*iterations of (3.1) with probability at least $1 - \delta$.*

*Proof.* Unrolling the recurrence from Lemma B.4,

$$\kappa^{(t)} \geq \kappa^{(0)} + \gamma C_1 \sum_{i=1}^{r} \sum_{s=0}^{t-1} \mu_i (\kappa^{(s)})^{i-1} \big(1 - (\kappa^{(s)})^2\big) - \gamma \left| \sum_{s=0}^{t-1} \nu^{(s)} \right|. \tag{B.21}$$

Since $\{\nu^{(s)}\}_{s=0}^{T-1}$ are independent mean-zero sub-Weibull random variables with $\Theta(1)$ tail parameter, we have, for some constant $C_3 > 0$,

$$\mathbb{E}\left[\left|\sum_{s=0}^{T-1} \nu^{2s}\right|^2\right] = \sum_{s=0}^{T-1} \mathbb{E}\big[|\nu^{2s}|^2\big] \leq C_3 T. \tag{B.22}$$

Moreover,

$$\mathbb{P}\left(\max_{0 \leq t \leq T-1} \left|\sum_{s=0}^{t} \nu^{2s}\right|^2 \geq 4C_3 \delta^{-1} T\right) \leq \frac{\delta}{4C_3 T} \mathbb{E}\left[\max_{0 \leq t \leq T-1} \left|\sum_{s=0}^{t} \nu^{2s}\right|^2\right] \quad \text{by Markov's inequality}$$

$$\leq \frac{\delta}{C_3 T} \mathbb{E}\left[\left|\sum_{s=0}^{T-1} \nu^{2s}\right|^2\right] \quad \text{by Doob's maximal inequality.} \tag{B.23}$$

Assume without loss of generality that $\kappa \geq 2d^{-1/2}$. Our bound on the dynamics after $t$ updates becomes, with probability at least $1 - \delta$,

$$\kappa^{(t)} \geq 2d^{-1/2} + \gamma C_1 \sum_{i=1}^{r} \sum_{s=0}^{t-1} \mu_i (\kappa^{(s)})^{i-1} (1 - (\kappa^{(s)})^2) - 2\gamma C_3^{1/2} \delta^{-1/2} T^{1/2}$$

$$= 2d^{-1/2} + \gamma C_1 \sum_{i=1}^{r} \sum_{s=0}^{t-1} \mu_i (\kappa^{(s)})^{i-1} (1 - (\kappa^{(s)})^2) - \gamma^{1/2} C_4 \delta^{-1/2} \min_{\substack{1 \leq i \leq r \\ \mu_i > 0}} \mu_i^{-1/2} d^{-(\frac{i-2}{4} \vee 0)} \tag{B.24}$$

for some $C_4 = \tilde{\Theta}(1)$ by definition of $T$. Now, recalling that $\gamma \leq C\delta \max_{1 \leq i \leq r} \mu_i d^{-(\frac{i}{2} \vee 1)}$, we have

$$\gamma^{1/2} C_4 \delta^{-1/2} \min_{\substack{1 \leq i \leq r \\ \mu_i > 0}} \mu_i^{-(\frac{i-2}{4} \vee 0)} \leq C^{1/2} \left(\min_{\substack{1 \leq i \leq r \\ \mu_i > 0}} \mu_i^{-\frac{1}{2}} d^{-(\frac{i-2}{4} \vee 0)}\right) \max_{1 \leq i \leq r} \mu_i^{1/2} d^{-(\frac{i}{4} \vee \frac{1}{2})}$$

$$\leq C^{1/2} C_4 d^{-1/2}, \tag{B.25}$$

which can be made less than $d^{-1/2}$ by choosing $C$ sufficiently small. Hence, our final (high probability) upper bound for the multi-step dynamics is

$$\kappa^{(t)} \geq d^{-1/2} + \gamma C_1 \sum_{i=1}^{r} \sum_{s=0}^{t-1} \mu_i (\kappa^{(s)})^{i-1} \big(1 - (\kappa^{(s)})^2\big). \tag{B.26}$$

Now, we unroll each of the $r$ terms in the expected dynamics and determine how quickly each one reaches $c \asymp 1/\operatorname{polylog} d$. Consider terms where $\mu_i > 0$. Since $\kappa^{(t)} \lesssim 1/\operatorname{polylog} d$ by assumption,

we may absorb the factor $(1 - (\kappa^{(t)})^2)$ into the constant $C_1$ (abusing notation). On the other hand, the contributions of terms with $\mu_i \leq 0$ will be negligible by Assumption 3.2.

For the $i = 1$ term, the noiseless dynamics give

$$
\begin{aligned}
& d^{-1/2} + \gamma C_1 \mu_i t \geq 2c \\
\iff \ & t \geq \gamma^{-1} C_1^{-1} \mu_i^{-1} (2c - d^{-1/2}) = \Theta(\gamma^{-1} \mu_i^{-1}).
\end{aligned}
\tag{B.27}
$$

When $i = 2$, we have, by Grönwall's Inequality (Lemma A.3)

$$
\begin{aligned}
& d^{-1/2} + \gamma C_1 \mu_i \sum_{s=0}^{t-1} \kappa^{(s)} \geq d^{-1/2} (1 + \gamma C_1 \mu_i)^t \geq 2c \\
\iff \ & t \log (1 + \gamma C_1 \mu_i) \geq \log 2c + \frac{1}{2} \log d \\
\iff \ & t \geq \frac{\log 2c + \frac{1}{2} \log d}{\log(1 + \gamma C_1 \mu_i)} = \tilde{\Theta}(\gamma^{-1} \mu_i^{-1}),
\end{aligned}
\tag{B.28}
$$

where the final equality follows from the fact $x - \frac{x^2}{2} \leq \log(1 + x) \leq x$ for $x \in (0, 1)$.

Lastly, for $i \geq 3$, we have, from the Bihari-LaSalle inequality (Lemma A.4),

$$
\begin{aligned}
& d^{-1/2} + \gamma C_1 \mu_i \sum_{s=0}^{t-1} (\kappa^{(s)})^{i-1} \geq \frac{d^{-1/2}}{(1 - \frac{1}{2} \gamma C_1 \mu_i d^{-\frac{i-2}{2}} t)^{\frac{1}{i-2}}} \geq 2c \\
\iff \ & d^{-\frac{i-2}{2}} \geq (2c)^{i-2} - \frac{1}{2} (2c)^{i-2} \gamma C_1 \mu_i d^{-\frac{i-2}{2}} t \\
\iff \ & t \geq 2\gamma^{-1} C_1^{-1} \mu_i^{-1} d^{\frac{i-2}{2}} ((2c)^{i-2} - d^{-\frac{i-2}{2}}) = \Theta(\gamma^{-1} \mu_i^{-1} d^{\frac{i-2}{2}}).
\end{aligned}
\tag{B.29}
$$

Thus, the maximum weak recovery time is indeed

$$
T = \min_{\substack{1 \leq i \leq r \\ \mu_i > 0}} \tilde{\Theta}(\gamma^{-1} \mu_i^{-1} d^{\frac{i-2}{2} \vee 0}).
\tag{B.30}
$$

$\square$

## B.5 Sample Complexity Lower Bound

The proof of the matching sample complexity lower bound proceeds much in the same way as that of the upper bound.

**Proposition B.6** (Generic Sample Complexity Lower Bound). *Fix $c \gtrsim 1/\operatorname{polylog} d$. Suppose $\langle \boldsymbol{\theta}_*, \boldsymbol{w}^{(0)} \rangle \lesssim d^{-1/2}$. Then, there exists a constant $\tilde{C} \gtrsim 1/\operatorname{polylog} d$ such that for all $\delta \in (0, 1)$, setting $\gamma \leq \tilde{C} \delta \max_{1 \leq i \leq r} \mu_i d^{-(\frac{i}{2} \vee 1)}$ gives $\langle \boldsymbol{\theta}_*, \boldsymbol{w}^{(t)} \rangle < c$ for all iterations $t \leq T$ of (3.1), where*

$$
T(\eta) = \min_{\substack{1 \leq i \leq r \\ \mu_i > 0}} \tilde{\Theta}(\gamma^{-1} (\mu_i(\eta))^{-1} d^{\frac{i-2}{2} \vee 0}),
\tag{B.31}
$$

*with probability at least $1 - \delta$.*

*Proof of Proposition B.6.* The projection error is trivial to handle, as we obtain

$$
\kappa^{(t+1)} = \frac{\kappa^{(t)} + \gamma \langle \boldsymbol{\theta}_*, \boldsymbol{g}^{(t)} \rangle}{\|\boldsymbol{w}^{(t)} + \gamma \boldsymbol{g}^{(t)}\|} \leq \kappa^{(t)} + \gamma \langle \boldsymbol{\theta}_*, \boldsymbol{g}^{(t)} \rangle,
\tag{B.32}
$$

since $\langle \boldsymbol{w}^{(t)}, \boldsymbol{g}^{(t)} \rangle = 0$ and $\|\boldsymbol{w}\| = 1$. Moreover, from Section B.3, the expected one-step update to the alignment is given by

$$
\mathbb{E}[\langle \boldsymbol{\theta}_*, \boldsymbol{g} \rangle] = \sum_{k=1}^{r} k! \mu_k \langle \boldsymbol{\theta}_*, \boldsymbol{w} \rangle^{k-1} (1 - \langle \boldsymbol{\theta}_*, \boldsymbol{w} \rangle^2).
\tag{B.33}
$$

Therefore, with the initialization $\kappa^{(0)} \leq \tilde{C}_0 d^{-1/2}$ for a positive constant $\tilde{C}_0$, the full dynamics are

$$
\begin{aligned}
\kappa^{(t+1)} &\leq \kappa^{(t)} + \gamma \sum_{i=1}^{r} i! \mu_i (\kappa^{(t)})^{i-1} (1 - (\kappa^{(t)})^2) + \gamma \nu^{(t)} \\
&\leq \kappa^{(0)} + \gamma \sum_{i=1}^{r} \sum_{s=0}^{t-1} i! \mu_i (\kappa^{(s)})^{i-1} (1 - (\kappa^{(s)})^2) + \gamma \left| \sum_{s=0}^{t-1} \nu^{(s)} \right| \\
&\leq \tilde{C}_0 d^{-1/2} + \gamma \sum_{i=1}^{r} \sum_{s=0}^{t-1} i! \mu_i (\kappa^{(s)})^{i-1} (1 - (\kappa^{(s)})^2) + \gamma C_3^{1/2} \delta^{-1/2} T^{1/2} \\
&\leq 2\tilde{C}_0 d^{-1/2} + \gamma \sum_{i=1}^{r} \sum_{s=0}^{t-1} i! \mu_i (\kappa^{(s)})^{i-1} (1 - (\kappa^{(s)})^2),
\end{aligned}
\tag{B.34}
$$

where the third line follows from the martingale bound in the previous subsection, and the last line follows from the constraint $\gamma \leq \tilde{C} \delta \max_{1 \leq i \leq r} \mu_i d^{-(\frac{i}{2} \vee 1)}$ with $\tilde{C}$ taken sufficiently small. Then, finding the minimum weak recovery time proceeds exactly as in the previous section, using Grönwall's inequality for $i = 2$ and the Bihari-LaSalle inequality for $i \geq 3$, once again giving

$$
T = \min_{\substack{1 \leq i \leq r \\ \mu_i > 0}} \tilde{\Theta} \left( \gamma^{-1} \mu_i^{-1} d^{\frac{i-2}{2} \vee 0} \right).
\tag{B.35}
$$

$\square$

Together, the sample complexity upper and lower bounds imply Theorem 3.3.

## B.6 Strong Recovery

Now, starting with $w$ that has achieved weak recovery, we characterize the maximum number $T'$ of subsequent updates of the form (3.1) required to achieve strong recovery with high probability.

**Proposition B.7** (Strong Recovery Given Weak Recovery). *Let $\varepsilon > 0$. Suppose that $\langle \theta_*, w^{(0)} \rangle \geq 2c$ for some $c \gtrsim 1/\operatorname{polylog} d$. Then, there exists a constant $C > 0$ such that for all $\delta \in (0, 1)$, setting $\gamma \leq C \delta d^{-1} \varepsilon \max_{1 \leq i \leq r} \mu_i c^{i-1}$ implies that the update rule (3.1) achieves $\langle \theta_*, w \rangle \geq 1 - \varepsilon$ within*

$$
T' = \min_{\substack{1 \leq i \leq r \\ \mu_i > 0}} \tilde{\Theta} (\gamma^{-1} \varepsilon^{-1} \mu_i^{-1})
\tag{B.36}
$$

*iterations with probability at least $1 - \delta$.*

**Remark B.8.** *If $\varepsilon = \tilde{\Theta}(1)$, then $T' \lesssim T$. That is, achieving weak recovery is the bottleneck during training.*

**Remark B.9.** *In the algorithms we consider in Section 4, we have $\max_{1 \leq i \leq r} \mu_i = \Theta(1)$. Therefore, given that weak recovery has already been achieved, then strong recovery for $\varepsilon = \tilde{\Theta}(1)$ proceeds after at most $\tilde{\Theta}(d)$ additional iterations with high probability when $\gamma \asymp d^{-1}$.*

*Proof.* Similarly to Section B.3, we have a lower bound on the one-step dynamics:

$$
\kappa^{(t+1)} \geq \kappa^{(t)} + \gamma \sum_{i=1}^{r} i! \mu_i (\kappa^{(t)})^{i-1} (1 - (\kappa^{(t)})^2) + \gamma \nu^{(t)} - C_2 \gamma^2 d.
\tag{B.37}
$$

Since $\langle \theta, w^{(t)} \rangle \leq 1 - \varepsilon$ and Assumption 3.2 holds, we can re-write this as

$$
\kappa^{(t+1)} \geq \kappa^{(t)} + \gamma C_1 \varepsilon \sum_{i=1}^{r} \mu_i (\kappa^{(t)})^{i-1} + \gamma \nu^{(t)} - C_2 \gamma^2 d
\tag{B.38}
$$

for some constant $C_1 > 0$. Setting $\gamma \leq C \delta d^{-1} \varepsilon$ leads to

$$
C_2 \gamma^2 \kappa^{(t)} d \leq C_2 C \delta \varepsilon \gamma \max_{1 \leq i \leq r} \mu_i c^{i-1} \leq C_2 C \gamma \delta \varepsilon \max_{1 \leq i \leq r} \mu_i (\kappa^{(t)})^{i-1}.
\tag{B.39}
$$

Thus, taking $C$ sufficiently small ensures that this is a fraction of the dominant term in the population update.

Unrolling this over $t$ steps, we obtain, with probability at least $1 - \delta$,

$$
\begin{aligned}
\kappa^{(t)} &\geq 2c + \gamma C_1 \varepsilon \sum_{i=1}^{r} \sum_{s=0}^{t-1} \mu_i (\kappa^{(s)})^{i-1} - \gamma \left| \sum_{s=0}^{t-1} \nu^{(s)} \right| \\
&\geq 2c + \gamma C_1 \varepsilon \sum_{i=1}^{r} \sum_{s=0}^{t-1} \mu_i (\kappa^{(s)})^{i-1} - \gamma C_3^{1/2} \delta^{-1/2} T' \\
&= 2c + \gamma C_1 \varepsilon \sum_{i=1}^{r} \sum_{s=0}^{t-1} \mu_i (\kappa^{(s)})^{i-1} - \gamma^{1/2} C_4 \delta^{-1/2} \varepsilon^{-1/2} \min_{\substack{1 \leq i \leq r \\ \mu_i > 0}} \mu_i^{-1/2}
\end{aligned}
\tag{B.40}
$$

using the same martingale bound as in Section B.4. Now recalling $\gamma \leq C \delta d^{-1} \varepsilon \max_{1 \leq i \leq r} \mu_i c^{i-1}$, we have

$$
\gamma^{1/2} C_4 \delta^{-1/2} \varepsilon^{-1/2} \min_{\substack{1 \leq i \leq r \\ \mu_i > 0}} \mu_i^{-1/2} \leq C^{1/2} C_4 d^{-1/2},
\tag{B.41}
$$

which is of lower order than $c$. Hence, our final (high probability) upper bound for the multi-step dynamics is

$$
\kappa^{(t)} \geq c + \gamma C_1 \varepsilon \sum_{i=1}^{r} \sum_{s=0}^{t-1} \mu_i (\kappa^{(s)})^{i-1}.
\tag{B.42}
$$

We analyze how quickly this exceeds $1 - \varepsilon$. For the $i = 1$ term, we obtain

$$
\begin{aligned}
\kappa^{(t)} &\geq c + \gamma C_1 \varepsilon \mu_1 t \geq 1 - \varepsilon \\
&\iff t \geq (1 - \varepsilon - c) C_1^{-1} \gamma^{-1} \varepsilon^{-1} \mu_1^{-1} = \Theta(\gamma^{-1} \varepsilon^{-1} \mu_1^{-1}).
\end{aligned}
\tag{B.43}
$$

For the $i = 2$ term, we have, by Grönwall's inequality

$$
\begin{aligned}
\kappa^{(t)} &\geq c + \gamma C_1 \varepsilon \mu_2 \sum_{s=0}^{t-1} \kappa^{(s)} \geq c(1 + \gamma C_1 \varepsilon \mu_2)^t \geq 1 - \varepsilon \\
&\iff t \geq \frac{\log(1 - \varepsilon) - \log c}{\log(1 + \gamma C_1 \varepsilon \mu_2)} = \tilde{\Theta}(\gamma^{-1} \varepsilon^{-1} \mu_2^{-1}).
\end{aligned}
\tag{B.44}
$$

For the $i \geq 3$ terms, we have, by the Bihari-LaSalle inequality,

$$
\begin{aligned}
\kappa^{(t)} &\geq c + \gamma C_1 \varepsilon \mu_i \sum_{s=0}^{t-1} \left( \kappa^{(s)} \right)^{i-1} \geq \frac{c}{(1 - \frac{1}{2} \gamma C_1 \varepsilon \mu_i c^{i-2} t)^{\frac{1}{i-2}}} \geq 1 - \varepsilon \\
&\iff t \geq 2 \left( 1 - \left( \tfrac{c}{1-\varepsilon} \right)^{i-2} \right) \gamma^{-1} C_1^{-1} \varepsilon^{-1} \mu_i^{-1} c^{-(i-2)} = \Theta(\gamma^{-1} \varepsilon^{-1} \mu_i^{-1}).
\end{aligned}
\tag{B.45}
$$

Hence, the (high probability) maximum strong recovery time given weak recovery is indeed

$$
T' = \min_{\substack{1 \leq i \leq r \\ \mu_i > 0}} \tilde{\Theta}(\gamma^{-1} \varepsilon^{-1} \mu_i^{-1}).
\tag{B.46}
$$

$\square$

## B.7 Ridge Regression on the Second Layer

For completeness, we state the following result from [LOSW24] that outlines the sample complexity of proceeding from strong recovery to approximation of the target to arbitrary accuracy. In particular, if the error tolerance is of constant order, then the sample complexity obtaining of strong recovery (starting from weak recovery) is strictly larger.

**Proposition B.10** (Second Layer Training [LOSW24, Lemma 20]). *Let $\varepsilon > 0$ and $N = \tilde{\Theta}(\varepsilon^{-1})$. Suppose that $\tilde{\Theta}(N)$ neurons in (2.2) satisfy $\langle \boldsymbol{\theta}_*, \boldsymbol{w}_j \rangle \geq 1 - \varepsilon$. Let $b_j \sim \mathrm{Unif}([-C_b, C_b])$ such that*

$C_b = \tilde{O}(1)$. *Then, there exists a choice of penalty parameter $\lambda = \tilde{\Theta}(1)$ such that the solution $\hat{\boldsymbol{a}} = (\hat{a}_1, \dots, \hat{a}_N)$ of ridge regression with $\tilde{\Theta}(N^{-4} + \varepsilon^{-4})$ samples satisfies*

$$\mathbb{E}_{\boldsymbol{x} \sim \mathcal{N}(\boldsymbol{0}, \boldsymbol{I}_d)}\left[\left|\left| \frac{1}{N} \sum_{j=1}^{N} \hat{a}_j \sigma_j(\langle \boldsymbol{x}, \boldsymbol{w}_j\rangle + b_j) - \sigma_*(\langle \boldsymbol{x}, \boldsymbol{\theta}_*\rangle) \right|\right|^2 \right] \lesssim \varepsilon^2. \tag{B.47}$$

*with high probability.*

The key assumption in the above is that a constant proportion (up to polylogarithmic factors) of the neurons achieve strong recovery. Recall that the initial alignment is sufficiently large with constant order probability (Section B.1), and that each of weak (Section B.4) and strong (Section B.6) recovery occur with high probability given such an initialization.

# C   SGD Variants

Recall from our proof of Theorem 3.3 in the previous section that the coefficients

$$\mu_i := \mathop{\mathbb{E}}_{(a,b) \sim \mathcal{N}(\boldsymbol{0}, \boldsymbol{I}_2)} \mathbb{E}_\zeta \left[ \psi_\eta(\sigma_*(a) + \zeta, b) \mathsf{He}_i(a) \mathsf{He}_{i-1}(b) \right], \quad i \in [r]. \tag{C.1}$$

are the key quantities governing the sample complexity of an online algorithm that fits in our framework. We detail the computation of these coefficients for each of the three SGD variants we consider in this work: online SGD (Section 4.1), alternating SGD (Section 4.3), and batch reuse SGD (Section 4.2). This along with Theorem 3.3 immediately imply the corollaries in Section 4 on the sample complexity of these algorithms. Additionally, in Section C.4, we investigate how alternating SGD can be generalized to an online algorithm for a $D$-layer neural network and calculate the $\mu_i$.

## C.1   Online SGD

As discussed in Section 4.1, given a fresh data point $(\boldsymbol{x}, y)$ the spherical online SGD update has the form

$$\boldsymbol{w}^{(t+1)} \leftarrow \boldsymbol{w}^{(t)} + \gamma y \sigma'(\langle \boldsymbol{x}, \boldsymbol{w}\rangle) \boldsymbol{P}_{\boldsymbol{w}^{(t)}}^\perp \boldsymbol{x}, \quad \boldsymbol{w}^{(t+1)} \leftarrow \frac{\boldsymbol{w}^{(t+1)}}{||\boldsymbol{w}^{(t+1)}||}. \tag{C.2}$$

Therefore, under our general framework introduced in Section 3, the update oracle is

$$\psi_\eta(y, z) = y \sigma'(z). \tag{C.3}$$

Hence, for $i \in [r]$,

$$\mu_i = \mathbb{E}_\zeta \mathbb{E}_{a,b}[(\sigma_*(a) + \zeta)\sigma'(b) \mathsf{He}_i(a) \mathsf{He}_{i-1}(b)] = u_i(\sigma_*) u_{i-1}(\sigma') = i u_i(\sigma) u_i(\sigma_*). \tag{C.4}$$

## C.2   Batch Reuse SGD

The update for Batch Reuse SGD (Algorithm 1) takes the form

$$\tilde{\boldsymbol{w}}^{(t)} \leftarrow \boldsymbol{w}^{(t)} + \eta y \sigma'(\langle \boldsymbol{x}, \boldsymbol{w}^{(t)}\rangle) \boldsymbol{P}_{\boldsymbol{w}^{(t)}} \boldsymbol{x}, \quad \boldsymbol{w}^{(t+1)} \leftarrow \frac{\boldsymbol{w}^{(t)} + \gamma y \sigma'(\langle \boldsymbol{x}, \tilde{\boldsymbol{w}}\rangle) \boldsymbol{P}_{\boldsymbol{w}^{(t)}}^\perp \boldsymbol{x}}{||\boldsymbol{w}^{(t)} + \gamma y \sigma'(\langle \boldsymbol{x}, \tilde{\boldsymbol{w}}\rangle) \boldsymbol{P}_{\boldsymbol{w}^{(t)}}^\perp \boldsymbol{x}||}. \tag{C.5}$$

Combining the two steps (and disregarding normalization for the time being), we have

$$\boldsymbol{w}^{(t+1)} = \boldsymbol{w}^{(t)} + \gamma y \sigma'\big(\langle \boldsymbol{x}, \boldsymbol{w}^{(t)}\rangle + \eta y \sigma'(\langle \boldsymbol{x}, \boldsymbol{w}^{(t)}\rangle)\langle \boldsymbol{x}, \boldsymbol{P}_{\boldsymbol{w}^{(t)}}^\perp \boldsymbol{x}\rangle\big) \boldsymbol{P}_{\boldsymbol{w}^{(t)}}^\perp \boldsymbol{x} \tag{C.6}$$

The presence of $||\boldsymbol{x}||_{\boldsymbol{P}_{\boldsymbol{w}^{(t)}}^\perp}^2$ in the update prevents us from immediately casting this into our formalism. We handle this as follows. Using a Taylor expansion, we have

$$\boldsymbol{w}^{(t+1)} = \boldsymbol{w}^{(t)} + \gamma y \sum_{k=1}^{r} \frac{\sigma^{(k)}(\langle \boldsymbol{x}, \boldsymbol{w}^{(t)}\rangle) y^{k-1} \eta^{k-1} \sigma'(\langle \boldsymbol{x}, \boldsymbol{w}^{(t)}\rangle)^{k-1} ||\boldsymbol{x}||_{\boldsymbol{P}_{\boldsymbol{w}^{(t)}}^\perp}^{2(k-1)}}{(k-1)!}. \tag{C.7}$$

Note that $||\boldsymbol{x}||^2_{\boldsymbol{P}_{\boldsymbol{w}^{(t)}}} \sim \chi^2_{d-1}$ and therefore $\mathbb{E}[||\boldsymbol{x}||^{2(i-1)}_{\boldsymbol{P}_{\boldsymbol{w}^{(t)}}}] = \Theta(d^{i-1})$. This, along with the assumption $\eta d \lesssim 1$, allows us to replace $||\boldsymbol{x}||^{2(i-1)}_{\boldsymbol{P}^{(t)}}$ in each term of the Taylor expansion with $d^{i-1}$ and add a sub-Weibull remainder term $\xi^{(t)}$ with $O(1)$ tail parameter:

$$\boldsymbol{w}^{(t+1)} - \boldsymbol{w}^{(t)} \asymp \gamma y \sum_{k=1}^{r} \sigma^{(k)}(\langle \boldsymbol{x}, \boldsymbol{w}^{(t)}\rangle)y^{k-1}(\eta d)^{k-1}\big(\sigma'(\langle \boldsymbol{x}, \boldsymbol{w}^{(t)}\rangle)\big)^{k-1}\boldsymbol{P}_{\boldsymbol{w}^{(t)}}\boldsymbol{x} + \gamma\xi^{(t)}\boldsymbol{P}_{\boldsymbol{w}^{(t)}}\boldsymbol{x}.$$
(C.8)

The $\xi^{(t)}$ term can then be absorbed into the noise that appears in the multi-step analysis in Sections B.4, B.5, and B.6. Hence, we can take

$$\psi_\eta(y, z) = \sum_{k=1}^{r}(\eta d)^{k-1}\sigma^{(k)}(z)\big(\sigma'(z)\big)^{k-1}y^k.$$
(C.9)

And so, for $i \in [r]$,

$$
\begin{aligned}
\mu_i &= \sum_{k=1}^{r}(\eta d)^{k-1}\mathbb{E}_\zeta\mathbb{E}_{a,b}\big[(\sigma_*(a) + \zeta)^k\sigma^{(k)}(b)(\sigma'(b))^{k-1}\mathsf{He}_i(a)\mathsf{He}_{i-1}(b)\big] \\
&= \sum_{k=1}^{r}(\eta d)^{k-1}u_{i-1}(\sigma^{(k)}(\sigma')^{k-1})\mathbb{E}_\zeta\mathbb{E}_a\bigg[\sum_{l=0}^{k}\binom{k}{l}(\sigma_*(a))^l\zeta^{k-l}\mathsf{He}_i(a)\bigg] \\
&\asymp \sum_{k=1}^{r}(\eta d)^{k-1}u_{i-1}\big(\sigma^{(k)}(\sigma')^{k-1}\big)u_i(\sigma_*^k).
\end{aligned}
$$
(C.10)

## C.3  Alternating SGD

The alternating SGD (Algorithm 2) update for a single neuron is

$$\tilde{a}^{(t+1)} \leftarrow a + \eta y\sigma(\langle \boldsymbol{x}, \boldsymbol{w}^{(t)}\rangle), \quad \boldsymbol{w}^{(t+1)} \leftarrow \frac{\boldsymbol{w}^{(t)} + \gamma y\tilde{a}^{(t+1)}\sigma'(\langle \boldsymbol{x}, \boldsymbol{w}^{(t)}\rangle)}{||\boldsymbol{w}^{(t)} + \gamma y\tilde{a}^{(t+1)}\sigma'(\langle \boldsymbol{x}, \boldsymbol{w}^{(t)}\rangle)||}.$$
(C.11)

Note that we only use the second layer update $\tilde{a}$ in order to update the first layer parameters $\boldsymbol{w}$. We do not replace the second layer parameter with $\tilde{a}$ at the subsequent iteration, but instead keep $a$. For simplicity, assume $a = 1$. Then,

$$\psi_\eta(y, z) = y\sigma'(z) + \eta y^2\sigma(z)\sigma'(z).$$
(C.12)

Noticing that the first term in the update is the same as in the previous subsection, we have, for $i \in [r]$,

$$
\begin{aligned}
\mu_i &= iu_i(\sigma)u_i(\sigma_*) + \eta\mathbb{E}_\zeta\mathbb{E}_{a,b}\big[(\sigma_*(a) + \zeta)^2\sigma(b)\sigma'(b)\mathsf{He}_i(a)\mathsf{He}_{i-1}(b)\big] \\
&= iu_i(\sigma)u_i(\sigma_*) + \eta u_{i-1}(\sigma\sigma')\mathbb{E}_\zeta\mathbb{E}_a\big[(\sigma_*^2(a) + 2\zeta\sigma_*(a) + \zeta^2)\mathsf{He}_i(a)\big] \\
&= iu_i(\sigma)u_i(\sigma_*) + \eta u_{i-1}(\sigma\sigma')u_i(\sigma_*^2).
\end{aligned}
$$
(C.13)

## C.4  "Deep" Alternating SGD

We define a $D$-layer neural network student by the recurrence

$$f(\boldsymbol{x}) = f_{D-1}(\boldsymbol{x}), \quad f_0(\boldsymbol{x}) = \boldsymbol{W}\boldsymbol{x}, \quad f_i(\boldsymbol{x}) = \boldsymbol{A}_i\sigma(f_{i-1}(\boldsymbol{x})), i \in [D-1],$$
(C.14)

where $\boldsymbol{W}_0 \in \mathbb{R}^{N \times d}$ as before and $\boldsymbol{A}_i \in \mathbb{R}^{N_{i+1} \times N_i}$ such that $N_1 = N$ and $N_D = 1$. We are still interested in recovery of the ground truth direction $\boldsymbol{\theta}_*$ by the first-layer weights $\boldsymbol{W}$. To make the theoretical analysis tractable, we consider the simplified sparse network where $N_1 = N_2 = \cdots = N_{D-1} = N$, $\boldsymbol{A}_1$ is a $N_2 \times N_1$ matrix of ones[6], and $\boldsymbol{A}_2 = \boldsymbol{A}_3 = \cdots = \boldsymbol{A}_{D-1} = \boldsymbol{I}_N$ with off-diagonal entries frozen at zero during training (i.e., they do not receive gradients). This prevents

---

[6]Note that we chose the same initialization for our two-layer network in the previous subsection.

interactions between weights that would render the analysis intractable. Hence, the output of the network is of the form

$$f(\boldsymbol{x}) = \sum_{j=1}^{N} a_j^{(D-1)} \sigma\big( \circ \cdots \circ \sigma(a_j^{(1)} \sigma(\langle \boldsymbol{x}, \boldsymbol{w}_j \rangle))\big). \tag{C.15}$$

Hence, to analyze weak recovery, it suffices to focus on a single summand (where we drop the subscript $j$ for convenience):

$$a^{(D-1)} \sigma\big( \circ \cdots \circ \sigma(a^{(1)} \sigma(\langle \boldsymbol{x}, \boldsymbol{w} \rangle))\big), \tag{C.16}$$

which we express as the recurrence

$$F(z) = F_{D-1}(z), \quad z = F_0(z) = \langle \boldsymbol{x}, \boldsymbol{w} \rangle, \quad F_i(z) = a^{(i)} \sigma\big(F_{i-1}(z)\big), i \in [D-1]. \tag{C.17}$$

We propose the following update rule inspired by our alternating SGD algorithm:

$$z^{(t)} \leftarrow \langle \boldsymbol{x}, \boldsymbol{w}^{(t)} \rangle$$

$$\tilde{a}^{(i)} \leftarrow a^{(i)} + \eta y \bigg( \prod_{j=i+1}^{D-1} a^{(i)} \sigma'(F_{j-1}(z^{(t)})) \bigg) \sigma\big(F_{i-1}(z^{(t)})\big), \quad i \in [D-1]$$

$$\boldsymbol{w}^{(t+1)} \leftarrow \boldsymbol{w}^{(t)} + \gamma y \bigg( \prod_{i=1}^{D-1} \tilde{a}^{(i)} \sigma'\big(F_{i-1}(z^{(t)})\big) \bigg) \boldsymbol{P}_{\boldsymbol{w}^{(t)}}^{\perp} \boldsymbol{x}, \quad \boldsymbol{w}^{(t+1)} \leftarrow \frac{\boldsymbol{w}^{(t+1)}}{||\boldsymbol{w}^{(t+1)}||}. \tag{C.18}$$

Expanding the update for $\boldsymbol{w}$ (before normalization) gives

$$\boldsymbol{w}^{(t+1)} = \boldsymbol{w} + \gamma y \prod_{i=1}^{D-1} \bigg[ \bigg( a^{(i)} + \eta y \bigg( \prod_{j=i+1}^{D-1} a^{(j)} \sigma'(F_{j-1}(z)) \bigg) \sigma(F_{i-1}(z)) \bigg) \sigma'(F_{i-1}(z)) \bigg] \boldsymbol{P}_{\boldsymbol{w}}^{\perp} \boldsymbol{x}, \tag{C.19}$$

where we have omitted the superscript $(t)$ on the right-hand side for readability. This fits into our framework (3.1) since the $a^{(i)}$ remain constant. In fact, for simplicity, we may fix all $a_i = 1$ for all $i \in [D-1]$. Our update oracle is then

$$\psi_\eta(y, z) = \sum_{i=0}^{D-1} \bigg[ \eta^i y^{i+1} \sum_{S \in \mathcal{P}_i([D-1])} \bigg( \prod_{j \notin S} \sigma'(\sigma^{\circ(j-1)}(z)) \bigg)$$

$$\cdot \bigg( \prod_{k \in S} \bigg( \prod_{l=k+1}^{D-1} \sigma'(\sigma^{\circ(l-1)}(z)) \bigg) \sigma^{\circ k}(z) \sigma'(\sigma^{\circ(k-1)}(z)) \bigg) \bigg], \tag{C.20}$$

where $\mathcal{P}_i([D-1])$ denotes the set of all subsets of $[D-1]$ of cardinality $i$. Now, assuming that the Hermite coefficients of the relevant compositions and products of $\sigma$ and $\sigma'$ are positive, this gives

$$\mu_i \asymp \sum_{j=1}^{D} \eta^{j-1} u_i(\sigma_*^j). \tag{C.21}$$

Under an optimal choice of $\gamma$, Theorem 3.3 implies a sample complexity of

$$T = \max_{1 \le i \le D} \tilde{\Theta}\big(\eta^{-2(i-1)} d^{(p_i-1)\vee 1}\big) \tag{C.22}$$

for deep alternating SGD to attain weak recovery.

Now, the positivity of the relevant Hermite coefficients is a nontrivial assumption. We consider the special case where $\sigma(z) = z^2$. Then, the update will take the form

$$\psi_\eta(y, z) = C \sum_{i=0}^{D-1} \bigg[ \eta^i y^{i+1} \sum_{S \in \mathcal{P}_i([D-1])} \bigg( \prod_{j \notin S} z^{2(j-1)\vee 1} \bigg) \prod_{k \in S} \prod_{l=k+1}^{D-1} z^{2(l-1)} z^{2k} z^{2(k-1)\vee 1} \bigg) \bigg]. \tag{C.23}$$

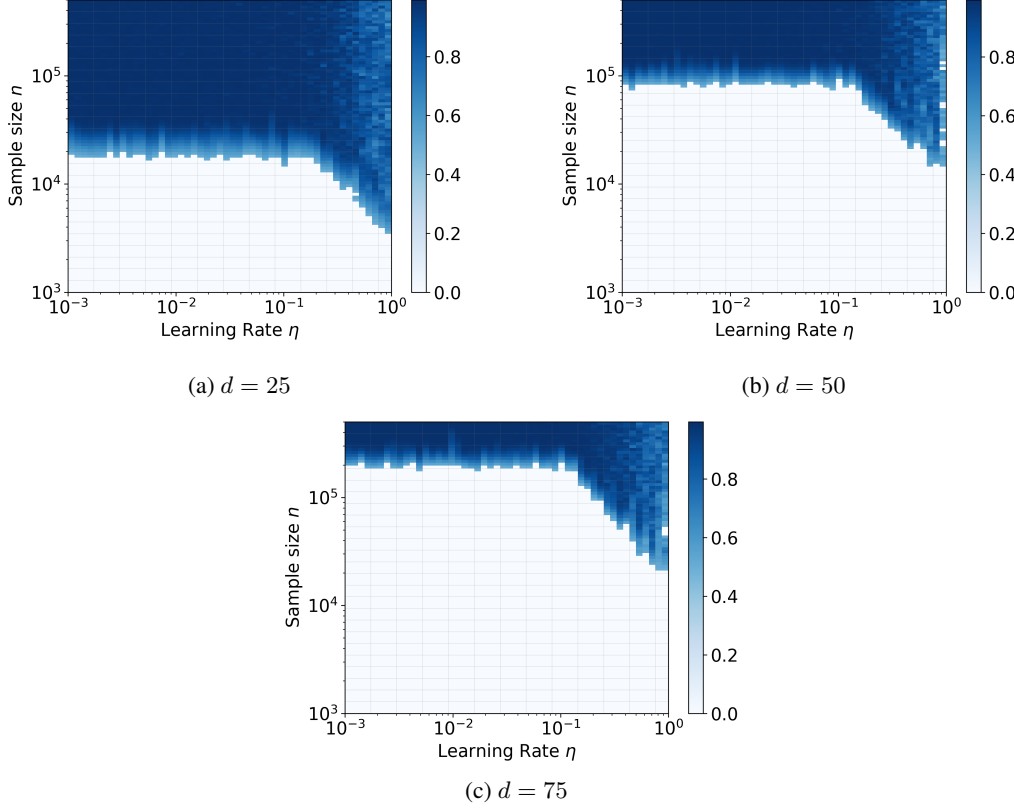

(a) $d = 25$

(b) $d = 50$

(c) $d = 75$

Figure 2: Alignments $\langle \boldsymbol{w}, \boldsymbol{\theta}_* \rangle$ greater than 0.5 for alternating SGD with different choices of $\eta$ and $n$. The hyperparameter $\gamma$ is chosen according to Corollary 4.3. Results are averaged over 10 runs.

for some positive constant $C$. Every term in the sum above contains an odd power of $z$ (this is most easily seen by separately considering the cases $1 \in S$ and $1 \notin S$). Hence, the Hermite coefficient $\mathbb{E}_{z \sim \mathcal{N}(0,1)}[\psi_\eta(y, z)\mathsf{He}_{i-1}(z)]$ is zero for $i$ odd and positive for $i$ even. In particular, $\mu_i$ is zero for $i$ odd and has the form (C.21) for $i$ even.

It is immediate that weak recovery is achieved with $\tilde{\Theta}(d)$ complexity if $\eta = \tilde{\Theta}(1)$ if $u_2(\sigma_*^j) > 0$ for some $j \in [D]$ and $u_2(\sigma_*^k) \geq 0$ for all $j \neq k$. This is a strict improvement over alternating SGD on a two-layer neural network when $p$ and $p_2$ are larger than 2 but $p_j = 2$ for some $j > 2$ *and* over batch reuse SGD under the same conditions and quadratic $\sigma$.

The above has the limitation that it will not recover in $\tilde{\Theta}(d)$ time if $u_2(\sigma_*^k) = 0$ for all $k \in [D]$ but $u_1(\sigma_*^k) > 0$ for at least one such $k$. For $p_3 = 1$, this can be resolved by taking $\sigma(z) = z^3$ and $D = 3$, in which case $u_0([\sigma'(\sigma(z))]^2\sigma(\sigma(z))\sigma(z)\sigma'(z)) > 0$ and hence $\mu_1 > 0$. However, we cannot assume to know the target a priori, and a more generally applicable choice of $\sigma$ is preferable (perhaps a randomized approach as in [LOSW24]). We leave a more thorough examination of potential choices of activation function to future work.

## D    Experiment Details

In this section, we provide the details on the experiment that generated Figure 1 in the main text and discuss additional experiments on batch reuse SGD and online SGD in the same setting. Code for all experiments is available online.[7] Throughout, we consider a noiseless single-index teacher (2.1) with $\sigma_* = \mathsf{He}_3$, $\boldsymbol{\theta}_* = \boldsymbol{e}_1$, and a two-layer neural network student (2.2) with $N = 1$ hidden neuron and no bias, i.e., $f(\boldsymbol{x}) = a\sigma(\langle \boldsymbol{x}, \boldsymbol{w} \rangle)$. The network is initialized with $a = 1$, the first entry of $\boldsymbol{w}$ equal

---

[7] https://github.com/kctsiolis/inf2genexp

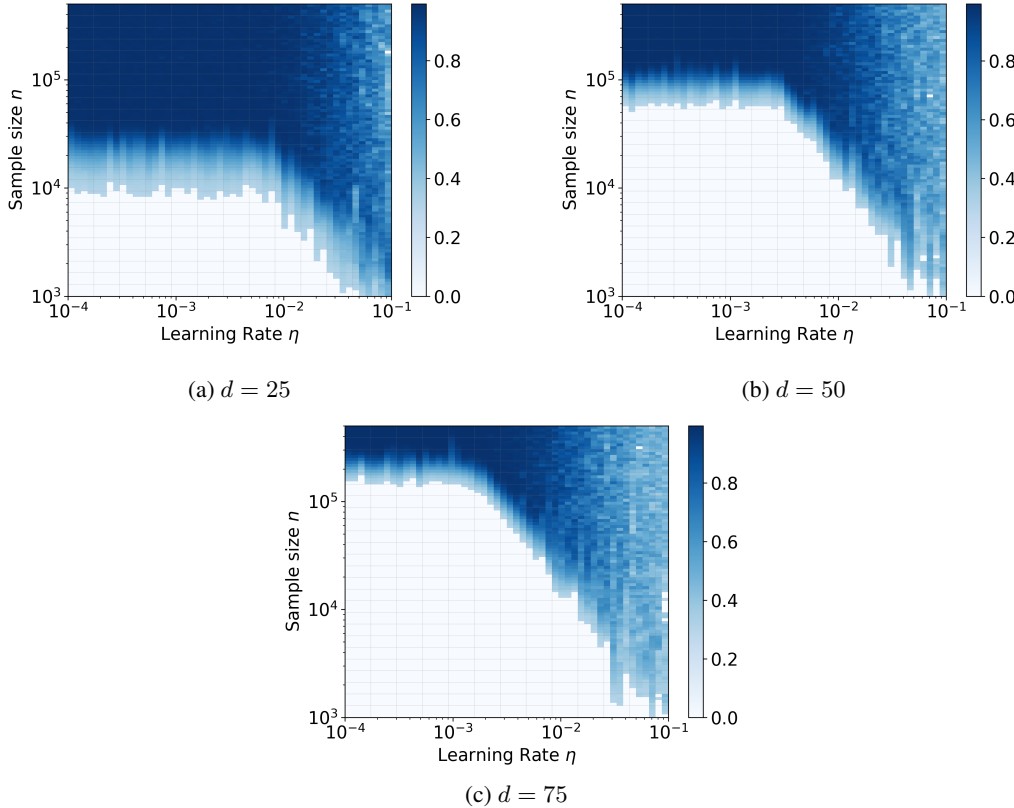

(a) $d = 25$

(b) $d = 50$

(c) $d = 75$

Figure 3: Alignments $\langle w, \theta_* \rangle$ greater than 0.5 for batch reuse SGD with different choices of $\eta$ and $n$. The hyperparameter $\gamma$ is chosen according to Corollary 4.2. Results are averaged over 10 runs.

to $1/\sqrt{d}$, and the remaining entries of $w$ are drawn from the uniform distribution over the sphere $\mathbb{S}^{d-2}(\sqrt{1 - 1/d})$. This ensures that for each simulation we start with the same initial alignment so that there is a fair comparison across learning rates.

For all algorithms, we experiment with $d \in \{25, 50, 75\}$ and take a logarithmically spaced mesh of 50 learning rate values. We consider the range $\eta \in [10^{-3}, 1]$ for alternating SGD, $\eta \in [10^{-4}, 10^{-1}]$ for batch reuse SGD, and $\gamma \in [10^{-4}, 1]$ for online SGD. We choose $\gamma = \max\{d^{-3/2}, \eta d^{-1}\}$ as per Corollary 4.3 (disregarding constants), $\gamma = \max\{d^{-3/2}, \eta\}$ for batch reuse SGD as per Corollary 4.2, and $\eta = 0$ for online SGD. We empirically verify the theoretically predicted phase transition in $\eta$ for alternating and batch reuse SGD and the predicted consistent decrease of the sample complexity with $\gamma$ for online SGD.

Each of the algorithms is implemented exactly as specified in Section 4, including the projection and normalization steps. We train in a single-pass over the data (i.e., one epoch) with fixed batch size $B = 128$. In other words, we perform $\lfloor n/B \rfloor$ online updates. Collecting the alignments $\langle w, \theta_* \rangle$ for each $(\mathrm{lr}, n)$ combination in our mesh gives the colorbars in Figures 1,3,4. To more clearly visualize recovery of $\theta_*$ and the phase transitions for alternating and batch reuse SGD, we threshold alignments at 0.5.

For alternating SGD (Figure 2), we observe that the sample complexity remains flat for small $\eta$ before decaying as power law after reaching a critical value, as predicted by Corollary 4.3. Also in line with our predictions is the observation that this critical value decreases with $d$. (Note that for this example, Corollary 4.3 predicts the phase transition occurs at $\eta \asymp d^{-1/2}$.) The results for batch reuse SGD (Figure 3) are similar, with the only difference being that the phase transition occurs at a smaller critical value of $\eta$ than it does for alternating SGD. This is is in line with the prediction of Corollary 4.2, which predicts the phase transition at $\eta \asymp d^{-3/2}$. Lastly, for online SGD, we observe that the

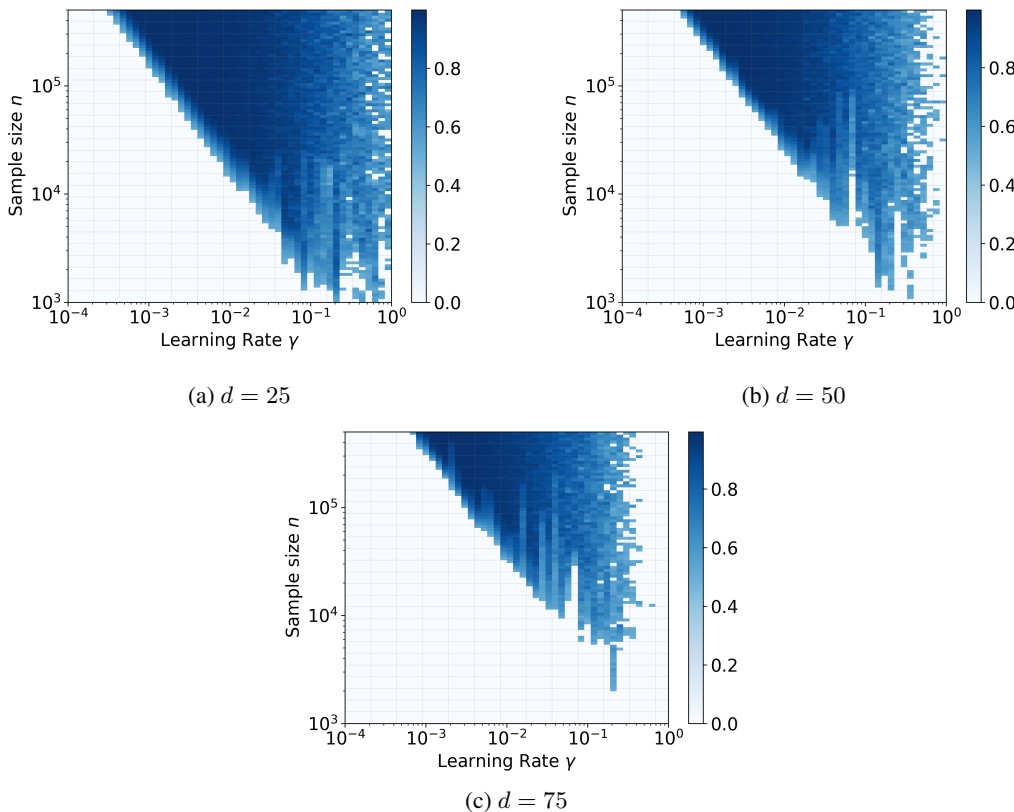

(a) $d = 25$

(b) $d = 50$

(c) $d = 75$

Figure 4: Alignments $\langle \boldsymbol{w}, \boldsymbol{\theta}_* \rangle$ greater than 0.5 for online SGD with different choices of $\gamma$ and $n$. Note that $\eta = 0$ in this case. Results are averaged over 10 runs.

sample complexity decays as power law in $\gamma$, as expected from Theorem 3.3. We also note that when $\gamma$ is chosen too large, the training becomes unstable and we no longer consistently recover.

