# OpenReview forum: "From Information to Generative Exponent: Learning Rate Induces Phase Transitions in SGD"
_NeurIPS.cc/2025/Conference — NeurIPS 2025 poster_

### Official Review · Reviewer_atEW · 2025-06-13

**Clarity:** 4
**Significance:** 3
**Originality:** 3
**Rating:** 5
**Confidence:** 4

**Summary:**

The study builds upon a series of works concerning sample complexity estimates for the high-dimensional limit of single index models. The authors consider the case of a two-layer neural network trained by algorithms belonging to a class of optimization methods based on generalized versions of the gradient of the MSE loss $\psi_\eta$, where $\eta$ is an additional learning rate parameter. The main theorem establishes a formula for the sample complexity, which depends on the learning rates $\gamma$ and $\eta$, as well as on the Hermite coefficients of the student's and teacher's link functions.

The authors argue that the novelty lies in the generality of the algorithmic class to which the theorem applies. They then show that both _vanilla SGD_  and  _batch reuse SGD_ belong to this class, and they establish the relationship between the new estimates and the information and generative exponents. They demonstrate how, by tuning $\eta$, it is possible to transition from a regime where learning is governed by the information exponent to one where learning is faster and governed by the generative exponent.

In addition, they introduce a new variant of SGD, termed _Alternating SGD_, and show how it can improve standard SGD when $\eta$ is sufficiently large. This variant can be further enhanced when applied to deeper networks.

**Questions:**

- About figure 1, I would strongly suggest to plot results for different values of $d$, to show how the phenomenology portrayed passes to the high-dimensional limit. Should we see a sharp vertical transition between blue and white region for $d\to \infty$? It may also help to continue the plot for even larger values of $\eta$.
- does your analysis offer practical insigths for tuning learning rate in networks, comments on this and simulations on more realistic and complex task may be interesting to practitioners.

### Questions and comments:
- In Theorem 3.3 you need to require an upper bound on $\gamma$. What happens for larger values? From the proof sketch, it seems that it is not possible to ignore the normalization term of the iterate expansion. Is that the tight value for such constant, beyond which the dynamics become too noisy?
- About Figure 1, I would strongly suggest plotting results for different values of $d$, to show how the portrayed phenomenology extends to the high-dimensional limit. Should we see a sharp vertical transition between the blue and white regions for $d \to \infty$? It may also help to extend the plot to even larger values of $\eta$.
- Does your analysis offer practical insights for tuning learning rates in networks? Comments on this aspect, together with simulations on more realistic and complex tasks, may be useful to practitioners.
- Can you clarify exactly what are your contributions in the batch reuse SGD setting?
- Can you comment on whether a non-polynomial $\sigma$ introduces just technical difficulties, or could give rise to different phenomenologies?
- The deep network case discussed in Appendix C.3 seems to be one of the strongest contributions of the paper, showing how depth can enhance learning. In particular, the final discussion in Appendix C.3 seems relevant, concerning which properties of the Hermite coefficients of $\sigma$ and $\sigma_*$ would make the deep framework better or worse than multi-pass SGD. You make comments only about the three-layer case; can they be generalized to larger $D$?
- Can your analysis be coupled with the smoothing techniques from [DNGL23] to reach the optimal sample complexity thresholds?

### Suggestion
From the communication point of view, in my opinion there is a bit too much emphasis on the fact that this work establishes the importance of the learning rate. This point is stressed both in the abstract, introduction, and conclusions. For instance, the conclusions section starts with:

> This work demonstrates that the learning rate is a fundamental factor in determining the sample complexity of gradient-based algorithms for learning single-index models with neural networks.

However, I think most readers already know that the learning rate is an impactful factor in determining the sample complexity in any learning problem. It is a core aspect of gradient methods, and most of the referenced works on single index models have detailed discussions about learning rates. I believe that the excessive emphasis on this aspect could risk drawing attention away from other important contributions of the paper. Moreover, it is even technically a bit misleading, since the usual SGD learning rate $\gamma$ is treated similarly to previous work, and it is the additional parameter $\eta$ that is the true focus of this work.

In my opinion, the merits of this work lie in the definition of a general framework that allows studying and finding optimal learning rates for a broad class of gradient-based optimization methods, providing a unified framework in which information and generative exponents appear as limit cases. I would suggest emphasizing more the generality and versatility of these results, and how the additional learning rate $\eta$ introduced in this novel class of learning algorithms can impact the sample complexity.

**Ethical Concerns:**

["NO or VERY MINOR ethics concerns only"]

**Final Justification:**

The authors addressed all my questions, especially the key (and for me puzzling) limitation to polynomial activation functions, explaining that they will extend to a much larger class of activation functions. This is why I increase my rating to 5.

**Limitations:**

The authors address the main limitations of the paper.

**Paper Formatting Concerns:**

No concerns

**Quality:**

4

**Strengths And Weaknesses:**

### Strenghts:
- well written and clear.
- theorem 3.3 is a general results provides a unified framework in which generative and information exponents appear as limit cases.
- the class of algorithm to which theorem 3.3 applies seems to be very broad, and Alternating SGD seems to be a nice example that could be of interest for practitioners
- the sketch of the proof insection 5 manages to convey clearly the key idea of the argument, showing how all the terms in the Hermite expansion of the Loss can become relevant depending on the scaling of $\eta$.
- the technical appendices contain thorough proofs of the main results of the paper, and even rigorous analyses that go beyond the main focus of the paper, such as appendix C.3 that examines the case of a deeper network. Or appendix B.6 and B.7 that provide already known, but necessary context on how to rigorously bridge the gap between weak and strong recovery.

### Weaknesses:
- cumbersome notation for hermite coefficients, especially formula 3.2, which is one of the central definitions in the paper. Writing the definition in terms of the integral over 2 independent gaussian variables, instead of this complicated notation with two nested $\to$, would have been shorter. I understand that in the paper you need to Hermite expand different functions, and need a compact notation for that. Nevertheless I suggest to provide the explicit intregrals at least once for the important formulas.
- since the area of single index model has now many research works that investigate similar questions of sample complexity, it is hard to assess which contributions are original, and which overlap with previous work. I do not see much novelty on the side of the analysis of _batch reuse SGD_, however the overall analysis and _alternating SGD_ are completely new, to the best of my knowledge (even though possible overlaps with the very recent [CLW+25] could be investigated more).
- just polynomial link functions is a rather restrictive assumption, I believe simulations that test the phenomenology on non-polynomial link functions should be performed.
- there are no comments and simulations that test the proposed _alternating SGD_ in more realistic settings.

---

> ### Author Rebuttal · Authors · 2025-07-30
>
> We thank the reviewer for their feedback. Please find our response below.
>
> **Notation for $\mu(\eta)$:** We agree with the reviewer's suggestion to improve readability. In the final version, we will define $\mu_i(\eta) := \mathbb{E}[\psi_{\eta}(\sigma_*(a),b) He_i(a) He_{i-1}(b)]$, where $a,b$ are independent standard normal random variables.
>
> **Clarification on Contributions:** Our core technical contribution is a careful analysis of different SGD variants to learn single-index models, while keeping track of dependencies on the learning rate, achieved through a unified set of tools by looking at a general update rule. Through this analysis, we are able to characterize how the *query class and complexity of SGD-type algorithms change depending on the learning rate*, which is our main conceptual contribution. These contributions are different from those of [3] on the technical side by a precise analysis that keeps track of the learning rate and on the conceptual side as we cover regimes in between CSQ and SQ.
>
> As the reviewer mentioned, in the specific case of batch reuse SGD, our analysis reduces to that of [1], with the exception of explicit tracking of learning rate. However, to our knowledge, the fact that *alternating SGD* can also modify the query class of SGD is entirely novel.
>
> **Non-Polynomial Link Functions:** We can in fact admit any link function that has at most polynomial growth (as in [2]) without affecting our main result. We will modify Assumption 3.1 in the final version to indicate this. This is a much larger function class that includes ReLU and its variants, splines, as well as any bounded link function such as the logistic and probit. Assuming at most polynomial growth for $\sigma_*$ ensures that $\langle \boldsymbol{\theta}_*, \boldsymbol{g} \rangle$ is sub-Weibull, allowing us to make concentration arguments.
>
> **Realistic Simulations of Alternating SGD:** We would like to highlight that we only study alternating SGD as a simplified proof of concept, as it provides a novel mechanism for modifying the statistical query type of SGD, one that comes from updates on the second layer rather than the first. In practice when both layers are trained simultaneously for a number of steps on the same batch, this effect will be combined with the effect of updates on the first layer alone, which is captured by Algorithm 2. Therefore alternating SGD here should be seen as a theoretical construction (similar to the batch reuse algorithm of prior works) to allow us to understand more practical settings, rather than a new optimization algorithm. We will better emphasize this fact in the paper.
>
> **Varying $d$ in Experiments:** We repeated the experiments in Figures 1 and 4 with $d = 25$ and $d = 75$. Taking $d$ larger is not feasible on our GPU due to the large number of samples (and thus running time) required. The theory predicts that the sample complexity will be "flat'' until a phase transition at $\eta \asymp d^{-1/2}$, after which it smoothly decreases until $\eta \asymp 1$. There will not be a sharp vertical transition as described by the reviewer. Our experimental results match this prediction. In particular, the threshold for the phase transition decreases as $d$ increases.
>
> In our experiments, taking larger values of $\eta$ than those appearing in Figure 1 leads to instability and poor generalization for both alternating and batch reuse SGD. We have a similar observation for all three algorithms when $\gamma$ is taken too large.
>
> We will add these additional experimental results to the final version.
>
> **Insights on Learning Rate Tuning:** We would like to stress that our main goal is not to find an optimal learning rate for practitioners, but to understand how learning rate can affect the oracle access/query model and the subsequent sample complexity of SGD. Nevertheless, we agree with the reviewer on the importance of being able to provide practical recommendations.
>
> The suggestion from our results is that learning rates $\gamma$, $\eta$ should be taken as large as possible (while still ensuring convergence) so as to avoid incurring unnecessary additional training cost. In particular, in the cases of alternating SGD and batch reuse SGD, the coefficients $\mu_i(\eta)$ are non-decreasing in $\eta$ (as long as Assumption 3.2 holds so that the relative Hermite coefficients are positive). In fact, for *any* algorithm where $\eta$ has the interpretation of a learning rate and thus appears in $\psi_{\eta}$ only as a scaling factor, this property will hold. It is therefore immediate that the practitioner should take $\eta$ as large as possible. From our condition $\psi_{\eta} = O(1)$ — which ensures control over the errors arising from normalization, SGD noise, and label noise — this means taking taking $\eta \asymp d^{-1}$ for batch reuse SGD (as in [1]) and $\eta \asymp 1$ for alternating SGD.
>
> **The Upper Bound on $\gamma$:** The reviewer's intuition is exactly correct. Taking $\gamma$ sufficiently small is necessary to ensure that the “informative” terms in the gradient update dominate the errors arising from both the normalization and the noise. The bound we provide on $\gamma$ is tight in the sense that it is the largest possible value for which these errors remain controlled. Moreover, as alluded to in an earlier answer, we observe in experiments that taking $\gamma$ larger than the theoretical threshold leads to instability.
>
> **Deep Networks:** We can generalize our example with $\sigma(z) = z^2$ to any depth. This is because, up to constants, $\psi_{\eta}(y,z)$ will always take the form $\sum_{i=0}^{D-1} \eta y^{i+1} z^{k_i}$, where all $k_i$ are odd. The odd Hermite coefficients of odd monomials are always positive, while their even Hermite coefficients are zero. Hence, we can recover in $\tilde{\Theta}(d)$ time if $\eta \asymp 1$ and at least one of $u_2(\sigma_*^k) > 0$ for $k \in [D]$. We will add a detailed discussion to illustrate this point in Appendix C.3.
>
> **SQ-Optimality through Smoothing:** Yes, it is possible to achieve the SQ lower bound through a technique inspired by the smoothing approach in [2]. The recent paper [3] shows that this can be done by aggregating the update $\psi(y,\langle \boldsymbol{x},\tilde{\boldsymbol{w}}\rangle)$ for a large number of random perturbations $\tilde{\boldsymbol{w}}$ of the current iterate $\boldsymbol{w}$ (see Algorithm 1 and Theorem 4.2 in [3] for details). This achieves the SQ lower bound whenever $\mu_{p_*}(\eta)$ is positive and of constant order (see Assumption 4.1(b) in [3], where $\mathbb{E}[\zeta_{s*}(y) \psi_{s*-1}(y)]$ is exactly $\mu_{p*}$ in our notation). This will be the case for batch reuse SGD whenever $\eta$ is sufficiently large (on the order $d^{-1}$), whereas for alternating SGD it will hold whenever $\mathrm{IE}(\sigma_*^2) = p_*$ and $\eta \asymp 1$.
>
> **Emphasis on Learning Rate:** We highly appreciate the reviewer's suggestion to better frame our contributions. In the final version, we will establish a clearer distinction between the hyperparameters $\gamma$ and $\eta$, emphasizing that only the latter yields phase transitions in sample complexity whereas the former affects the sample complexity in a smooth way (due to the $\gamma^{-1}$ factor appearing the rate in Theorem 3.3). We will also highlight that our framework allows for the study of other hyperparameter and design choices, such as the role of depth in the alternating SGD algorithm.
>
> We are happy to answer any further questions.
>
> [1] Jason D. Lee, Kazusato Oko, Taiji Suzuki, and Denny Wu. Neural network learns low-dimensional polynomials with SGD near the information-theoretic limit. NeurIPS 2024.
>
> [2] Alex Damian, Eshaan Nichani, Rong Ge, and Jason D Lee. Smoothing the landscape boosts the signal for SGD: Optimal sample complexity for learning single index models. NeurIPS 2023.
>
> [3] Siyu Chen, Beining Wu, Miao Lu, Zhuoran Yang, and Tianhao Wang. Can neural networks achieve optimal computational-statistical tradeoff? an analysis on single-index model. ICLR 2025.

---

> > ### Comment · Reviewer_atEW · 2025-08-02
> >
> > Thanks for the in-depth answer to all my questions. In particular, I believe the possibility to generalise the analysis to non-polynomial activation functions is a very important improvement that allows to connect the setup to more realistic settings. I will increase my rating.

---

### Official Review · Reviewer_wVyS · 2025-06-29

**Clarity:** 4
**Significance:** 2
**Originality:** 4
**Rating:** 4
**Confidence:** 4

**Summary:**

This papers investigate the sample complexity of training two-layer neural networks to weakly learn intrinsic features with data generated from Gaussian single-index models.

Specifically, the authors focus on a set of online gradient-based algorithms that can be expressed with a general gradient oracle $\psi_\eta(y,\langle x, w\rangle)$ , where $\eta$ is a hyperparameter that determines the non-linearity of the gradient query.

Then, they stated the tight sample complexity for this general algorithm with a measure of non-correlational in $\eta$. Using this main result, they provided the tight sample complexity for vanilla online SGD, alternating SGD and batch-reused SGD. For alternating SGD and batch-reused SGD, they revealed a novel phase-transition in the intrinsic learning rate $\eta.$

**Questions:**

1. In the perspective of exploration, the pseudo-step is used to find a better gradient near the current point and smoothing can be viewed as a similar technique (however smoothing is used in some other scenarios with higher generative exponent). In that case, the polarization parameter $\lambda$ (see e.g. [1,2]) may plays a role. Would it be possible to analyze the analogous phenomenon for $\lambda$?
2. Would it be possible to extend this analysis to the general link functions, instead of the polynomial links?
3. The main result essentially depends on $\gamma$ (the learning rate for the spherical gradient) and there is certain notion of the optimal choice of $\gamma.$ But in the result of [2], they only require the learning rate to be sufficiently large. Is the choice of $\gamma$ for the technical reason or it matters in whether the sample complexity is optimal?
4. This type of model typically relates to the SQ-problem, where I tend to think that closing the computational statistical gap is more important (see the second con). Does this learning-rate phase transition relate to some more practical phenomenon or has practical implications? (I think it’s very different to EoS which is more realistic). Or potentially your characteriaztion can provides some other insights (see the first question).

**Ethical Concerns:**

["NO or VERY MINOR ethics concerns only"]

**Limitations:**

See cons.

**Paper Formatting Concerns:**

No.

**Quality:**

3

**Strengths And Weaknesses:**

Pros:

1. The writing is generally clear and easy to follow. Proof sketch in the maintext is essential to capture the key components.
2. The framework of the general gradient oracle is theoretically convenient and the characterization of the non-correlational property with $\mu(\eta)$ is quite critical in capturing the hardness of the problem (in terms of $\eta$).
3. The sample complexity phase transition in learning rate is a novel perspective for the batch-reused SGD in single index models. Previous work all focus on the cases where the psudo-step is taken with sufficiently large but reasonable step size, but the sample complexity for the batch-reused SGD with smaller learning rate remains unclear.

Cons

1. The definition of $\mu(\eta)$ is a bit obsecure at first glance. It would be better if the author can preview some examples of $\psi_\eta$  in Section 3 and see how $\eta$  appears. Also it would be better if authors can briefly explain how this quantity reigns the sample complexity in the main text & discuss if similar notion appears in other literatures.
2. The result is novel by completing the whole picture of batch-reusing, but quite incremental in terms of matching the lower bound, as the prior works have fully settled the optimal algorithms and sample complexity of various online SGD in single-index models.

---

> ### Author Rebuttal · Authors · 2025-07-31
>
> We thank the reviewer for their feedback. Please find our response below.
>
> **Clarifying the $\mu(\eta)$ Notation:** The formalism of the generic update rule $\psi(y,\langle \boldsymbol{x}, \boldsymbol{w} \rangle)$ was introduced in [1]; our only modification is to track the explicit dependence on a hyperparameter $\eta$, which appears in SGD variants that employ label transformation, for example. In fact, the expression denoted $\mathbb{E}[\zeta_{s*}(y) \psi_{s*-1}(y)]$ in Assumption 4.1(b) of [1] is exactly $\mu_{p_*}$ in our notation. While the authors of [1] assume that the latter quantity is of constant order (which in our examples amounts to taking $\eta$ as large as possible), we consider the sample complexity for a full range of $\eta$ values. We will add a discussion on this effect in Section 3 where we introduce the formalism and the $\mu(\eta)$ notation. We will also preview how this formalism captures batch reuse SGD (as per the reviewer's suggestion) for easier readability.
>
> **Incremental Progress Towards Matching SQ Lower Bounds:** We would like to highlight that our goal is not to find the optimal sample complexity under some oracle assumption, but rather to provide a detailed characterization of the failure modes of existing SGD variants that break out of CSQ. Namely, we show that incorrect choice of hyperparameters governing the size of non-correlational updates can lead to sample complexity as poor as online SGD. Moreover, any algorithm that fits our framework (including our novel alternating SGD) and satisfies $\mu_{p_*} = \tilde{\Theta}(1)$ can be made SQ-optimal with the smoothing-inspired approach in [1].
>
> **1) Smoothing Hyperparameters:** We believe that it is possible to adapt our framework to capture landscape smoothing and treat the associated hyperparameter (called $\gamma$ in [1] and $\lambda$ in [2]) as our $\eta$. This boils down to casting the algorithm's update as $\psi(y,\langle \boldsymbol{x},\boldsymbol{w} \rangle)$ and computing the coefficients $\mu_i$. This is more intricate than the three examples we consider in our work due to the random perturbations of the weight vector $\boldsymbol{w}$. In particular, for the algorithm in [2], the gradient update takes the form $\frac{1}{\sqrt{1+\lambda^2}} y \mathbb{E} [\sigma’(\frac{\langle \boldsymbol{w}, \boldsymbol{x} \rangle + \lambda \langle \boldsymbol{\xi}, \boldsymbol{x} \rangle}{1 + \lambda^2})]\boldsymbol{x}$, where $\boldsymbol{\xi}$ is drawn from the uniform distribution on $\mathbb{S}^{d-1}$ conditioned on being orthogonal to $\boldsymbol{w}$. Note that the distribution of $\langle \boldsymbol{\xi}, \boldsymbol{x} \rangle$ depends on $||\boldsymbol{x}||$ in addition to $\langle \boldsymbol{x},\boldsymbol{w} \rangle$. We believe that the proof techniques from the appendices of [2] to handle this can be of use here.
>
> On this point, it is also worth mentioning that that Theorem 1 of [2] establishes that that landscape smoothing SGD has sample complexity $\tilde{O}(d^{p-1} \lambda^{-2(p-2)})$ for $\lambda \in [1,d^{1/4}]$. Hence, there is a smooth (i.e., no phase transitions) interpolation between the online SGD complexity $\tilde{O}(d^{p-1})$ and the CSQ lower bound as a function of $\lambda$.
>
> The situation is more complicated for the SQ-optimal algorithm of [1], where the analysis assumes that the polarization parameter $\gamma$ is fixed at its optimal value. This choice directly influences the number of samples required per step to ensure concentration of the empirical gradient (Proposition E.4 in [1]). Moreover, the expected alignment between the gradient and $\theta_*$ depends on $\gamma^{p_*-1}$ (Proposition E.3 in [1]). We leave it to future work to understand the behaviour for suboptimal $\gamma$ and attempt to cast the algorithm into our framework.
>
> *Please note: We think the references [1,2] are missing in the review.*
>
> **2) More General Link Functions:** We can in fact handle any link function that has at most polynomial growth (as in [3]). We will modify Assumption 3.1 in the final version to indicate this. This is a much larger function class that includes ReLU and its variants, splines, as well as any bounded link function such as the logistic and probit. Assuming at most polynomial growth for $\sigma_*$ ensures that $\langle \boldsymbol{\theta}_*, \boldsymbol{g} \rangle$ is sub-Weibull. Indeed, the class of sub-Weibull random variables includes sub-Gaussian and sub-exponential random variables and is closed under transformations with at most polynomial growth (up to changing the tail parameter). This condition on the noise ensures that we can make a concentration of measure argument to control the errors arising from label noise and the stochastic gradient.
>
> **3) The Choice of $\gamma$:** The hyperparameter $\gamma$ must on the order of its largest possible value as per Theorem 3.3 so that optimal sample complexity can be achieved. This constraint is necessary to control the errors in the update to $\langle \boldsymbol{\theta}_*,\boldsymbol{w} \rangle$ that arise due to normalization, stochastic gradient noise, and label noise (see proof sketch in Section 5). It is analogous to the constraint placed on the hyperparameters in batch reuse SGD (see Theorem 2 in [3], Theorem 1 in [4]).
>
> **4) The Implications of the Phase Transition:** We had two main motivations to study the effect of learning rate on generalization:
>
> a) Our first motivation was to understand when gradient-based algorithms on squared loss can break out of CSQ. While prior works [3,4] focus on the role of batch reuse, we believe learning rate is another important aspect that determines the oracle class of gradient-based algorithms, as demonstrated by our examples.
>
> b) In practice, it has been observed that larger learning rate leads to better generalization [5]. There is no consensus on why this is the case, and works on edge of stability (EoS) are more concerned with the optimization dynamics than generalization behaviour. In that regard, we believe studies that link sample complexity and learning rate can be valuable. Specifically, our work shows that significant improvements in the generalization behaviour can occur *even when one is not operating in the EoS regime*. Further studies to better characterize practical settings where this is the case are interesting, but outside the scope of our current paper.
>
> We are happy to answer any further questions.
>
> [1] Siyu Chen, Beining Wu, Miao Lu, Zhuoran Yang, and Tianhao Wang. Can neural networks achieve optimal computational-statistical tradeoff? an analysis on single-index model. ICLR 2025.
>
> [2] Alex Damian, Eshaan Nichani, Rong Ge, and Jason D Lee. Smoothing the landscape boosts the signal for SGD: Optimal sample complexity for learning single index models. NeurIPS 2023.
>
> [3] Luca Arnaboldi, Yatin Dandi, Florent Krzakala, Luca Pesce, and Ludovic Stephan. Repetita iuvant: Data repetition allows SGD to learn high-dimensional multi-index functions. arXiv 2024.
>
> [4] Jason D. Lee, Kazusato Oko, Taiji Suzuki, and Denny Wu. Neural network learns low-dimensional polynomials with SGD near the information-theoretic limit. NeurIPS 2024.
>
> [5] Yuanzhi Li, Colin Wei, Tengyu Ma. Towards Explaining the Regularization Effect of Initial Large Learning Rate in Training Neural Networks. NeurIPS 2019.

---

> > ### Comment · Reviewer_wVyS · 2025-08-02
> >
> > Thanks for your in-depth response to my questions. I think they have been fully settled. I will keep my scores unchanged.

---

### Official Review · Reviewer_PnJT · 2025-07-02

**Clarity:** 3
**Significance:** 3
**Originality:** 3
**Rating:** 4
**Confidence:** 2

**Summary:**

The paper characterizes the relationship between the learning rate and sample complexity for a class of gradient-based algorithms including both correlational and non-correlational updates. A phase transition is observed when the learning rate increases from small to large. Examples include online SGD, alternating SGD, batch reuse SGD are discussed. Some very simple numerical experiments are included.

**Questions:**

(1) It would be great if you can discuss when Theorem 3.3. will lead to phase transitions. I understand that you have discussed several examples in Section 4, but it would also be interesting to discuss for the general setup as in Theorem 3.3, when you will have phase transitions, when you will not, depending on the properties of $\mu_{i}(\eta)$.

(2) Some references need to be cleaned up. For example, in [AVPVF23], sgd should be SGD, in [Cla87], gronwall should be Gronwall, in [CCM11], gaussian should be Gaussian.

**Ethical Concerns:**

["NO or VERY MINOR ethics concerns only"]

**Final Justification:**

I have now raised the score. The author(s) have addressed the issues I raised.

**Limitations:**

It seems the author(s) did not discuss the limitations in the paper.

**Paper Formatting Concerns:**

No.

**Quality:**

3

**Strengths And Weaknesses:**

Strengths:

(1) The main results are easy to follow, and the meaning of phase transition is clear in this context.

(2) Various examples are included in the class of gradient-based algorithms, and they are properly discussed.

(3) Figure 1 seems to be in line with what the theory suggests.

Weaknesses:

(1) The class of models considered in this paper still seem to be a bit special. It would be interesting to see if phase transitions can occur in more general settings.

(2) Figure 1 is based on a toy example. More numerical experiments will be helpful, especially in terms of whether the phase transitions claimed in the paper can be observed in various numerical settings in practice, which will also make the theory more convincing. For example Figure 1 illustrates the toy example for Corollary 4.2., but there is no numerical illustration for Corollary 4.3. for batch reuse SGD, which according to Figure 3, should lead to multiple phase transitions. It would be great to see numerically that it is indeed the case. For Corollary 4.1. for online SGD, the theory does not suggest there will be phase transitions, and it will also be useful to numerically check that indeed no phase transitions will be observed numerically.

---

> ### Author Rebuttal · Authors · 2025-07-31
>
> We thank the reviewer for their feedback. Please find our response below.
>
> **The Class of Models under Consideration:** We would like to point out that despite their simplicity compared to practice, the class of Gaussian single-index models provides a tractable formalism for the theoretical study of feature learning [1,2,3]. The assumptions inherent to this setting are crucial to establish rigorous sample complexity bounds for SGD. In the paper, we assume that the link function $\sigma_*$ is polynomial, which is rather restrictive. We can in fact **extend this to any link function with at most polynomial growth** and will modify Assumption 3.1 in the final version to reflect this. This broadens our theory to capture much more general link functions in statistics and machine learning, such as ReLU (and its variants), the logistic, and the probit.
>
> **Additional Experiments:** As per the reviewer's suggestion, we have run additional experiments for online and batch reuse SGD. To start, we focused on our existing example of $\sigma_* = \operatorname{He}\_3$ since $\operatorname{IE}(\sigma_*) = 3$ and $\operatorname{IE}(\sigma_*^2) = 2$. For online SGD, there is *no phase transition* with respect to the learning rate $\gamma$ and the test error decreases at approximately a $\Theta(\gamma^{-1})$ rate, *as expected from theory*. On the other hand, we performed a larger-scale version of the experiment in Figure 4 for batch reuse SGD and confirmed that this algorithm *does* exhibit a phase transition due to the parameter $\eta$. Since we are not allowed to share supplementary figures during the rebuttal, we can only include these experiments in the final version.
>
> We would also like to highlight the fact that *experiments in practical settings have already observed that learning rate can affect generalization (and thus sample complexity)*, see e.g. [6]. Our study can be one way to approach a theoretical justification of this observation.
>
> **A General Discussion of the Phase Transition in Section 3:** We thank the reviewer for this suggestion, which will indeed help the readers better parse the main result. We provide a brief discussion below, and will it to the final version of the paper in Section 3.
>
> For simplicity, assume that all $\mu_i$ are positive. Consider the key equation $T(\eta) = \min_{1 \leq i \leq r} \tilde{\Theta}(\gamma^{-1} \mu_i(\eta)^{-1} d^{\frac{i-2}{2} \vee 0})$. The sample complexity is determined by the smallest term, whose index we can identify by $\alpha(\eta) = \operatorname{argmin}_{1 \leq i \leq r} \tilde{\Theta}(\gamma^{-1} \mu_i(\eta)^{-1} d^{\frac{i-2}{2} \vee 0})$. We say that the sample complexity exhibits *phase transitions* when $\alpha$ changes with $\eta$. We can identify potential phase transitions by comparing $\mu_i^{-1} d^{\frac{i-2}{2}} \leq \mu_j^{-1} d^{\frac{j-2}{2}}$. In particular, at fixed $d$, if $\mu_i$ is growing faster with $\eta$ than $\mu_j$, then the $i$th term will be more important in determining sample complexity for larger $\eta$. This is the phenomenon we illustrate with alternating SGD and batch reuse SGD in Section 4.
>
> Please let us know if we can clarify this further.
>
> **Referencing Issues:** We thank the reviewer for pointing these issues out. We have addressed these for the final version.
>
> **Limitations of the Paper**: We agree with the reviewer that our discussion of limitations should be more explicit, and we will rectify this in the final version. Currently we are limited to the setting of Gaussian single-index models under the specific assumptions in Section 3 in order for the analysis to be theoretically tractable. In the conclusion, we list possible extensions that would broaden the scope of our theory (e.g., multi-index models [4], more general input distributions [5]).
>
> We hope our responses have addressed the reviewer's concerns, and are happy to answer any further questions.
>
> [1] Gerard Ben Arous, Reza Gheissari, and Aukosh Jagannath. Online stochastic gradient descent on non-convex losses from high-dimensional inference. JMLR 2021.
>
> [2] Jimmy Ba, Murat A Erdogdu, Taiji Suzuki, Zhichao Wang, Denny Wu, and Greg Yang. High-dimensional asymptotics of feature learning: How one gradient step improves the representation. NeurIPS 2022.
>
> [3] Jason D. Lee, Kazusato Oko, Taiji Suzuki, and Denny Wu. Neural network learns low-dimensional polynomials with SGD near the information-theoretic limit. NeurIPS 2024.
>
> [4] Emmanuel Abbe, Enric Boix Adsera, and Theodor Misiakiewicz. SGD learning on neural networks: Leap complexity and saddle-to-saddle dynamics. COLT 2023.
>
> [5] Nirmit Joshi, Hugo Koubbi, Theodor Misiakiewicz, and Nathan Srebro. Learning single-index models via harmonic decomposition. arXiv 2025.
>
> [6] Yuanzhi Li, Colin Wei, Tengyu Ma. Towards Explaining the Regularization Effect of Initial Large Learning Rate in Training Neural Networks. NeurIPS 2019.

---

> > ### Comment · Reviewer_PnJT · 2025-08-04
> > **response**
> >
> > Thanks for the detailed response. I do not have further comments at this stage. I will raise the score.

---

### Official Review · Reviewer_NSB9 · 2025-07-03

**Clarity:** 3
**Significance:** 3
**Originality:** 4
**Rating:** 5
**Confidence:** 4

**Summary:**

This paper analyzes how the learning rate affects the sample complexity for neural networks learning gaussian single-index models. It establishes a general framework to show that for many gradient-based algorithms, the learning rate induces a phase transition in performance. Specifically, a small learning rate confines the algorithm to a complexity determined by the "information exponent," while a sufficiently large learning rate allows it to achieve a better complexity determined by the "generative exponent" for polynomial link function.

The authors further apply the framework to online SGD, SGD with square label transformation, and batch-reusing, and demonstrate how the phase transition occurs in each case.

**Questions:**

1. On Determining the Learning Rate: Beyond the specific analysis in this paper, is there a more general principle or rule of thumb for setting the optimal learning rate, or is it fundamentally tied to the specific properties of the model?

2. On Extending to Broader Function Classes: The framework in (3.1) appears general enough to accommodate non-polynomial link functions. Could the current results be extended to such cases? What are the primary technical barriers to this extension? Would criteria similar to $\mu_i(\eta)$ still be applicable for analyzing the sample complexity in a non-polynomial setting?

**Ethical Concerns:**

["NO or VERY MINOR ethics concerns only"]

**Final Justification:**

A theoretically solid paper that might be of interest to the community, and the authors' rebuttal resolved my concerns

**Limitations:**

yes

**Quality:**

3

**Strengths And Weaknesses:**

**Strengths**

This paper presents a well-structured and technically rigorous analysis that builds upon the extensive recent literature on learning single-index models. Its key contribution is investigating how the learning rate dictates the sample complexity, a crucial aspect that previous works often simplified by assuming either a sufficiently large or small learning rate.

1. The key contribution of this work is its investigation into the influence of the learning rate on sample complexity. The authors demonstrate how adjusting the learning rate, including inner learning rates for batch-reusing steps, allows the sample complexity to interpolate between two critical bounds:

     - The information exponent-related bound of $d^{p-1}$

     - The SQ bound of $\tilde O(d)$ for polynomial functions.

2. If my understanding is correct, this research can be viewed as an extension of the work by Lee et al. (2024), providing a more nuanced understanding of how learning rate selection directly impacts the sample efficiency of learning in single-index models.

**Weaknesses**:

A potential limitation of the proposed framework is the cumbersome process required to determine the optimal learning rate. This selection is not only sensitive, with Remark 3.4 indicating that changes in $\mu_i(\eta)$ can alter the optimal rate by a polynomial factor in the dimension, but it also presupposes detailed knowledge of the interplay between the link and activation functions. This dependence on specific model information may impede the practical implementation of these algorithms.

Another possible issue is that the proposed algorithms in Section 4.2 are still a little bit artificial, but the batch-reusing part seems already more natural then the version in Lee et al. (2024). Meanwhile, I'm also aware this is a common issue in the existing literature.

[1] Jason D. Lee, Kazusato Oko, Taiji Suzuki, and Denny Wu. Neural network learns low-dimensional 442 polynomials with SGD near the information-theoretic limit. In The Thirty-eighth Annual Confer443 ence on Neural Information Processing Systems, 2024.

---

> ### Author Rebuttal · Authors · 2025-07-30
>
> We thank the reviewer for their feedback. Please find our response below.
>
> **The Optimal Learning Rate:** Indeed, the optimal $\eta$ depends on the form of the $\mu_i(\eta)$ in general. However, for the cases alternating SGD and batch reuse SGD, these coefficients are non-decreasing in $\eta$ (as long as Assumption 3.2 holds so that the relative Hermite coefficients are positive). In fact, for *any* algorithm where $\eta$ has the interpretation of a learning rate and thus appears in $\psi_{\eta}$ only as a scaling factor, this property will hold. It is therefore immediate that the practitioner should take $\eta$ as large as possible while still ensuring convergence. From our condition $\psi_{\eta} = O(1)$ — which ensures control over the errors arising from normalization, SGD noise, and label noise — this means taking $\eta \asymp d^{-1}$ for batch reuse SGD (as in [1]) and $\eta \asymp 1$ for alternating SGD.
>
> **The Link Functions under Consideration:** We can in fact handle any link function that has at most polynomial growth (as in [2]). We will modify Assumption 3.1 in the final version to indicate this. This is a much larger function class that includes ReLU and its variants, splines, as well as any bounded link function such as the logistic and probit. Assuming at most polynomial growth for $\sigma_*$ ensures that $\langle \boldsymbol{\theta}_*, \boldsymbol{g} \rangle$ is sub-Weibull. Indeed, the class of sub-Weibull random variables includes sub-Gaussian and sub-exponential random variables and is closed under transformations with at most polynomial growth (up to changing the tail parameter). This condition on the noise ensures that we can make a concentration of measure argument to control the errors arising from label noise and the stochastic gradient.
>
> **The Artificial Nature of Alternating SGD:** We thank the reviewer for raising this concern, which allows us to clarify the point of studying Algorithm 1. Although this algorithm is somewhat contrived when compared with practice, we believe that its study is useful from a theoretical perspective. We see it as proof of concept to demonstrate what mechanisms might be at play in neural network training so that they achieve near-optimal sample complexity. Specifically, it reveals another mechanism for obtaining benefits of data repetition — one that comes from updates on the second layer rather than first. In practice when both layers are trained simultaneously for a number of steps on the same batch, this effect will be combined with the effect of updates on the first layer alone, which is captured by Algorithm 2.
>
> We also highlight the connection between Algorithm 1 and the literature on studying the two-timescales optimization dynamics of neural nets. By studying this algorithm, we are able to link the discrepancy in the learning rates to sample complexity, which is a novel perspective in this literature. Once again, the ideal algorithm would take multiple steps on the same batch and observe the combined effect of Algorithms 1 and 2, but studying this idealized setting is out of reach for our current techniques.
>
> We will better highlight these points in the final version to motivate the study of Algorithm 1.
>
> We are happy to answer any further questions.
>
> [1] Jason D. Lee, Kazusato Oko, Taiji Suzuki, and Denny Wu. Neural network learns low-dimensional polynomials with SGD near the information-theoretic limit. NeurIPS 2024.
>
> [2] Luca Arnaboldi, Yatin Dandi, Florent Krzakala, Luca Pesce, and Ludovic Stephan. Repetita iuvant: Data repetition allows SGD to learn high-dimensional multi-index functions. arXiv 2024.

---

> > ### Comment · Reviewer_NSB9 · 2025-08-05
> >
> > Thanks for the authors' response. I do not have further questions. My original evaluation remains unchanged.

---

### Note · Authors · 2025-08-13

We thank the reviewers and the area chair for their time, their interest in our work, and their feedback that helped significantly improve our draft. We are encouraged that our contributions and follow-up discussions were appreciated by the reviewers, as reflected in the decision by Reviewers PnJT and atEW to increase their score and all reviewers now recommending acceptance. We summarize the key takeaways from the discussion and the improvements for our forthcoming final version below.

During the rebuttal, we addressed the main concerns:

- We clarified that our framework can cover any link function $\sigma_*$ with at most polynomial growth.

- On the question of optimal hyperparameter choice, we noted that Theorem 3.3 implies that $\gamma$, $\eta$ should be taken as large as possible while still ensuring convergence (in the typical case where the $\mu_i$ are nondecreasing in $\eta$).

- We emphasized the place of our work relative to recent literature on SQ-optimality: our paper illustrates that the query class and complexity of SGD-type algorithms — including those which can break out of CSQ — change depending on the learning rate.

- We highlighted that our framework is general enough to cover additional hyperparameters (beyond learning rate) in optimal algorithms, such as  those in landscape smoothing.

For the final version, we will:

- Include a larger suite of numerical simulations for each of the three example SGD variants in Section 4 — the results of which are consistent with our theory.

- Provide a more readable definition for the coefficients $\mu_i(\eta)$ and motivate it with an example earlier in the main text.

- Place greater emphasis in the main text on our ability to analyze alternating SGD on networks with more than two layers and show an extension of the quadratic activation example in Appendix C.3 to higher depths.

- Draw a clearer distinction between the roles of the hyperparameters $\gamma$ and $\eta$, emphasizing that, while both have a bearing on sample complexity, only the latter induces phase transitions.

- Add a general discussion of the phase transition immediately following the statement of Theorem 3.3.

We believe the suggestions from the discussion period will greatly enhance our paper, and we hope our contributions can more clearly highlight the impact of design choices on the complexity of SGD and its variants in high-dimensional settings.

---

### Decision · Program_Chairs · 2025-09-17

**Decision:**

Accept (poster)

**Comment:**

This paper studies how the learning rate influences the sample complexity of two-layer neural networks under the single-index model. The authors show that moving from small to large learning rates changes the query class and the sample complexity of SGD variants, determined by the information and generative exponents, respecticely.

The main contribution lies in a careful analysis of several SGD variants for learning single-index models, while explicitly tracking learning-rate dependencies. This is achieved through a unified framework based on a general update rule. The framework is further applied to online SGD, SGD with square label transformation, and batch-reusing, demonstrating phase transitions in each case.

The authors successfully addressed the reviewers’ concerns during the rebuttal, and all reviewers now recommend acceptance. However, several clarifications should be included in the final version: (i) a more explicit explanation of $\mu_i(\eta)$, (ii) a clear distinction between $\gamma$ and $\eta$, and (iii) further discussion on the choice of optimal learning rates, as also noted by Reviewer NSB9. In particular, comparisons to practical WSD learning rates and the possibility of a variational optimization over the function class of $\eta$ would strengthen the paper.